# Addressing Concept Mislabeling in Concept Bottleneck Models Through Preference Optimization

**Emiliano Penaloza** [1 2]   **Tianyue H. Zhang** [1 2]   **Laurent Charlin** [2 3 *]   **Mateo Espinosa Zarlenga** [4 *]

## Abstract

Concept Bottleneck Models (CBMs) propose to enhance the trustworthiness of AI systems by constraining their decisions on a set of human-understandable concepts. However, CBMs typically assume that datasets contains accurate concept labels—an assumption often violated in practice, which we show can significantly degrade performance (by 25% in some cases). To address this, we introduce the *Concept Preference Optimization* (CPO) objective, a new loss function based on Direct Preference Optimization, which effectively mitigates the negative impact of concept mislabeling on CBM performance. We provide an analysis on some key properties of the CPO objective showing it directly optimizes for the concept's posterior distribution, and contrast it against Binary Cross Entropy (BCE) where we show CPO is inherently less sensitive to concept noise. We empirically confirm our analysis finding that CPO consistently outperforms BCE in three real-world datasets with and without added label noise. We make our code available on Github[1].

## 1. Introduction

It is a well-known adage, etched in the memory of any computing student, that "garbage in" leads to "garbage out." Yet, when designing new machine learning (ML) methods whose success hinges on the availability of high-quality labeled data, this consideration is usually left undiscussed. We show that this oversight affects Concept Bottleneck Models (CBMs) (Koh et al., 2020), a popular but label-hungry family of interpretable neural architectures, when trained with mislabeled concepts. As a remedy, we propose a learning

---
[*]Equal Supervision [1]Université de Montreal [2] Mila - Québec AI Institute [3] HEC Montréal [4] University of Cambridge. Correspondence to: Emiliano Penaloza <emiliano.penaloza@mila.quebec>.

*Proceedings of the 42nd International Conference on Machine Learning*, Vancouver, Canada. PMLR 267, 2025. Copyright 2025 by the author(s).

[1]https://github.com/Emilianopp/ConceptPreferenceOptimization

objective that aids CBMs to learn tasks even in the presence of label noise.

CBMs have emerged as a promising solution to the opacity of traditional Deep Neural Networks (DNNs). By leveraging human-understandable concepts as intermediate representations during inference, CBMs can explain their predictions using high-level concepts (e.g., *"has tail"*, *"has whiskers"*, etc.) relevant to their downstream task (e.g., *"cat"*). This hierarchical formulation enables experts interacting with a CBM at test time to *intervene*, or correct, a mispredicted concept and trigger an update to the CBM's output prediction (Shin et al., 2023). With their intervenability (Marcinkevičs et al., 2024) and interpretability, CBMs are ideal model candidates for high-stakes tasks where verifiability is paramount.

Although promising, CBMs come with a significant constraint: their training requires the set of *correct* concept annotations for all samples. In practice, it is unrealistic to assume that potentially a hundred concepts per datum would be correctly labeled. As a point of reference, a recent study concludes that 12% of the ImageNet-1K animal validation images have an incorrect label (and "some classes having $> 90\%$ of images incorrect labels") (Luccioni & Rolnick, 2023). We might expect those percentages to be even higher for concept labels. Further, within real-world domains where CBMs are intended to be deployed, such as healthcare, datasets are inherently noisy (Sylolypavan et al., 2023) and plagued with subjective labels (Wei et al., 2024). Additionally, regardless of the correctness of the labeled data, the training pipeline of CBMs depends on data augmentations (e.g., random crops/flips, see Figure 1) that can obscure concepts. Such a training pipeline makes some concept mislabelling inevitable in CBMs, potentially affecting them even under optimal labels. Thus, developing CBMs that are robust to concept-label noise can significantly enhance their usability in real-world tasks (even when task labels are correct).

In this work, we take inspiration from the field of Preference Optimization (PO), which relaxes the assumption made in traditional supervised ML that their training data is sampled from the optimal data distribution. PO algorithms only assume *preference* of the preferred training labels, rather

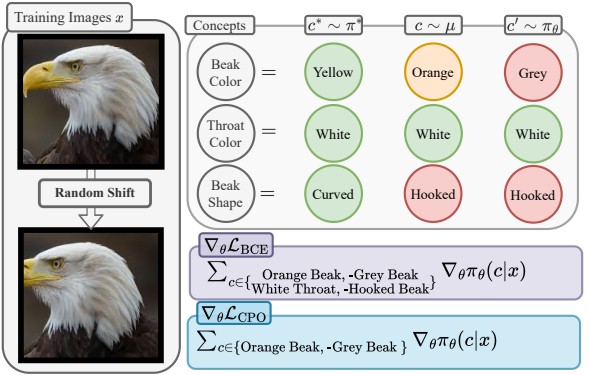

*Figure 1.* Visualization of the behavior of $\mathcal{L}_{\text{CPO}}$ and $\mathcal{L}_{\text{BCE}}$ under different scenarios, where the concepts are sampled from the optimal policy $\pi^*$, the empirical data $\mu$ and the amortized sampling policy $\pi_\theta$. In detail the scenarios for the concepts are: Beak color where the not-optimal, but not worst concept (orange) is preferred over incorrectly sampled (red). Throat color where sampled and empirical data align as optimal (green). Beak shape where both are incorrect. We analyze how this impacts $\mathcal{L}_{\text{BCE}}$ and $\mathcal{L}_{\text{CPO}}$ gradients: $\mathcal{L}_{\text{CPO}}$ may sample incorrect concepts, leading to no gradient updates and reduced sensitivity to noise, while $\mathcal{L}_{\text{BCE}}$ updates on all samples, regardless of correctness.

than their correctness, a relaxation that has been found to be particularly useful in noisy settings, such as recommender and information retrieval systems (Kaufmann et al., 2023; Bengs et al., 2021).

We propose *Concept Preference Optimization* (CPO), a policy optimization-inspired objective loss for CBMs. Figure 1 illustrates how CPO leverages pairwise comparisons of concept preferences to guide updates toward preferred concepts while mitigating the impact of incorrect gradients. Unlike traditional likelihood-based learning, which updates on all samples regardless of correctness, CPO selectively adjusts based on sampled preferences. This reduces sensitivity to noise by being able to mitigate incorrect gradient updates when incorrect concepts are sampled. Our analysis shows that CPO is equivalent to learning the posterior distribution over concepts, leading to more robust training. Empirically, we demonstrate that CPO not only improves CBM performance in noise-free settings but also significantly alleviates the impact of concept mislabeling.

## 2. Related Work

**Concept Learning (CL)** Concept Learning is a subfield of eXplainable AI (XAI) where models are designed to explain their prediction using human-understandable units of information, or *concepts* (Bau et al., 2017; Kim et al., 2018), that are relevant for a task of interest (Poeta et al., 2023). While CL methods use diverse approaches to produce concept-based explanations, most can be framed within

the context of a Concept Bottleneck Model (CBM) (Koh et al., 2020). A CBM is a neural architecture composed of (1) a *concept predictor* $\pi_\theta(c \mid x)$, which maps input features $x$ to a predicted distribution $c$ over a set of pre-defined concepts, and (2) a *label predictor* $f_\phi(c)$, which maps the set of predicted concepts $c$ to a downstream label $y$. By conditioning its task predictions on a set of concepts, CBMs can explain their prediction through their predicted concepts. They also allow for *concept interventions*, where, at test time, an expert interacting with the CBM can correct a handful of its mispredicted concepts leading to significant improvements in task accuracy (Shin et al., 2023).

Recent approaches have expanded the reach of CBMs across varying real-world setups. Concept Embedding Models (CEMs) (Espinosa Zarlenga et al., 2022) enhance the expressivity of concept representations to enable CBMs to be competitive in datasets with *incomplete* (Yeh et al., 2020) concept annotations. Post-hoc CBMs (Yuksekgonul et al., 2023), LaBOs (Yang et al., 2023), and Label-free CBMs (Oikarinen et al., 2023) instead address the difficulty of sourcing concept labels and retraining models by exploiting foundation and pretrained models. Further works improve the effectiveness of concept interventions by introducing new training losses (Espinosa Zarlenga et al., 2023), intervention policies (Chauhan et al., 2022), or considering inter-concept relationships (Havasi et al., 2022; Steinmann et al., 2023; Vandenhirtz et al., 2024; Raman et al., 2024).

Among these, the closest to our work are Probabilistic CBMs (ProbCBM) (Kim et al., 2023) and Stochastic CBMs (SCBMs) (Vandenhirtz et al., 2024). Both approaches frame CBMs probabilistically and learn to amortize the posterior distribution of an auxiliary latent variable between the concepts and the input data. ProbCBMs amortize the latent's posterior using a diagonal covariance matrix to estimate concept uncertainty. In contrast, SCBMs amortize the full covariance matrix and use it to estimate joint concept distributions for more efficient interventions. Both approaches have their benefits, but they both approximate the posterior of a latent variable and *not* the concept distributions. Conversely, we show how the CPO objective is equivalent to learning the posterior distribution of the concepts.

**Preference Optimization (PO)** PO is a powerful learning framework when the training labels are suboptimal, such as in recommender and information retrieval systems (Yue & Joachims, 2009; Shivaswamy & Joachims, 2012; Radlinski et al., 2008; Dudík et al., 2015). At their core, PO algorithms focus on learning a policy in setups where we lack an explicit reward signal but instead have access to relative preferences between pairs of labels – a weaker constraint. PO has become particularly important in the training of Large Language Models (LLMs) in the form of Reinforcement Learning from Human Feedback (RLHF), which is used

to guide policy optimization based on qualitative feedback (Ouyang et al., 2022b; team, 2024). While powerful, this approach is computationally expensive as the reward function and policy are trained separately. To alleviate this, Rafailov et al. (2023) introduce the *Direct Preference Optimization (DPO)* objective, which streamlines the process by jointly training both the reward function and policy.

Akin to traditional likelihood-based optimization approaches, DPO has the added benefit of being differentiable. In contrast to likelihood-based learning, however, DPO does not require a set of training labels sampled from the optimal data distribution, instead only assuming that a preference exists between any pair of labels (Bengs et al., 2021). Assuming optimal labels makes likelihood-based methods prone to overfitting to simple patterns (Arpit et al., 2017), making them vastly less robust to label noise (Goodfellow et al., 2016). Such a limitation is particularly relevant for traditional CBM training pipelines, which often include data augmentations and sample mislabels that can lead to concept labels differing from the optimal ones. To alleviate this, we extend the DPO objective to CBMs. We find that, as in language and retrieval tasks, training CBMs with our objective alleviates the effect of label noise.

## 3. Background

In this work, we approach learning CBMs through the use of PO and thus, adapt our notation accordingly.

**Concept Bottleneck Models** Concept Bottleneck Models (CBMs) (Koh et al., 2020) assume their training sets $(\mathbf{X}, \mathbf{C}, \mathbf{Y})$ are sampled i.i.d. from an empirical distribution $\mu(c, x, y)$, where $x \in \mathbf{X}$ are the input features, $c \in \mathbf{C}$ are the binary concepts labels $c = \{c_1, ..., c_k\}$, and $y \in \mathbf{Y}$ are the task labels. We assume that $\mu(c, x, y)$ may not necessarily be the same as $d^*(c, x, y)$, the *optimal data distribution* sampled from the optimal policy $\pi^*$. Specifically, we assume that they only differ at the concept level, meaning the empirical distribution's concept labels may be noisy while the task labels are always correct. We argue, however, that this difference is likely in practice as concept-specific noise may be accidentally added during training because of common data augmentations that may occlude concepts (e.g., random crops or shifts, see Figure 1). Moreover, concept-specific noise may naturally arise from subjective or fatigued labeling (Sylolypavan et al., 2023; Wei et al., 2024).

CBMs consist of two sub-models. First, a concept predictor, $\pi_\theta : \mathbf{X} \to \mathbf{C}^k$, maps the input $x$ onto an interpretable layer composed of predicted concepts, $\hat{c}$. The concept predictor is usually initialized using a pretrained image encoder $k_\theta$. Then, a task predictor, $f_\phi : \mathbf{C}^k \to \mathbf{Y}^m$, maps these predicted concepts to the task labels $\hat{y}$. In this work, we focus on *jointly trained CBMs*, which are trained end-to-end by minimizing the following objective weighted by a

hyperparameter $\lambda \in \mathbb{R}$:

$$\mathcal{L}_{\text{CBM}} = \mathcal{L}_{\text{CE}}(y, f_\phi(c)) + \lambda \mathcal{L}_{\text{BCE}}(c, \pi_\theta(c|x)).$$

The concept objective above optimizes the binary cross entropy (BCE) between the policy's predictions and the empirical data, which is known to be suboptimal under noisy settings (Goodfellow et al., 2016). Due this sensitivity, we take inspiration from modern PO algorithms, deriving a simple and computationally efficient objective that is equivalent to approximating the concept's posterior distribution and is more robust to noise compared to BCE.

**Direct Preference Optimization (DPO)** Traditionally, preference optimization using RLHF algorithms (Kaufmann et al., 2023) relies on learning a reward function through the Bradley-Terry preference model (Ouyang et al., 2022a; Kaufmann et al., 2023). Given a preference dataset $(c^w, c^l, x, y) \sim \mu^p$ one can learn a reward function capable of distinguishing preferred concepts $c^w$ from dispreferred ones $c^l$ by optimizing

$$\max_{r_\psi} \mathbb{E}_{(c^w, c^l, x) \sim \mu^p}[\log \sigma(r_\psi(c^w, x) - r_\psi(c^l, x))]$$

Where $r_\psi$ is a parameterized reward function learnt through the optimization process and $\sigma$ is the sigmoid function. Using this learned reward function, a policy can be trained with any RL algorithm. Most commonly employed is the proximal policy optimization (Schulman et al., 2017) algorithm, which imposes a KL constraint with a prior $\pi_0(c|x)$ on the standard reward maximization objective,

$$\max_{\pi_\theta} \mathbb{E}_{x \sim \mu, c \sim \pi_\theta}[r_\psi(x, c)] - \beta \mathbb{D}_{\text{KL}}(\pi_\theta(c|x) \| \pi_0(c|x)) \quad (1)$$

where $\beta$ is a hyperparameter controlling the prior's strength. However, this two-step procedure is computationally expensive and unstable. To address this, Rafailov et al. (2023) proposed the Direct Preference Optimization (DPO) algorithm. Showing that the optimal policy for this optimization problem can be expressed as

$$\pi^*(c|x) = \frac{1}{Z(x)} \pi_0(c|x) \exp\left(\frac{1}{\beta} r^*(x, c)\right) \quad (2)$$

where $Z(x) = \sum_c \pi_0(c|x) \exp\left(\frac{1}{\beta} r^*(x, c)\right)$ is the partition function, and $r^*(x, c)$ represents the optimal reward. Consequently, the optimal reward function can be expressed in terms of the optimal policy:

$$r^*(x, c) = \beta \log \frac{\pi^*(c|x)}{\pi_0(c|x)} + \beta \log Z(x) \quad (3)$$

Using this formulation, Equation 1 simplifies to

$$\max_{\pi_\theta} \mathbb{E}_{(c^w, c^l, x) \sim \mu} \left[\log \sigma\left(\log \frac{\pi_\theta(c^w|x)}{\pi_0(c^w|x)} - \log \frac{\pi_\theta(c^l|x)}{\pi_0(c^l|x)}\right)\right] \quad (4)$$

which is an offline objective that jointly trains the policy and reward functions.

# 4. Preference Optimization for CBMs

Although one could directly optimize the objective in Equation 4, doing so for CBMs would require a labeled dataset where preferences in concepts are explicitly specified. To circumvent this issue, we can leverage the empirical dataset and state its preference over a concept set sampled from $\pi_\theta$. The preference over a pair of concepts should hold specifically early on in training, where the policy is suboptimal compared to the empirical data. Throughout the rest of this section, we formally describe this algorithm, showing some key similarities and differences between it and $\mathcal{L}_{\text{BCE}}$.

## 4.1. Concept Bottleneck Preference Optimization

To leverage DPO to learn $\pi_\theta$, we re-formalize it as an online learning algorithm. We collect negative concept sets by sampling from the policy conditioned on the input features $c' \sim \pi_\theta(c|x)$.[2] We can then compare these negatively sampled concept sets with those sampled from the empirical data $c \sim \mu$, where we assume that the empirical set is *preferred* over the sampled set (i.e., $c \succ c'$). Note that this is a weaker assumption than that of traditional CBMs, as we only assume a preference over $c$ rather than its correctness. Using this, we introduce the *Concept Preference Optimization (CPO)* objective, an online formulation of Equation 4:

$$\mathcal{L}_{\text{CPO}} = -\mathbb{E}_{\substack{(x,c)\sim\mu \\ c'\sim\pi_\theta}} \left[ \log \sigma \left( \log \frac{\pi_\theta(c|x)}{\pi_0(c|x)} - \log \frac{\pi_\theta(c'|x)}{\pi_0(c'|x)} \right) \right].$$
(5)

When used in language modeling, $\pi_0$ is defined as the model after a supervised fine-tuning procedure. Here, we train the model from scratch. In practice, we could impose a prior on the concept labels which relate to either the input or the task label e.g., $\pi_0(c|x,y)$. We briefly explore such applications in § 5.4, but otherwise assume a uniform prior unless otherwise stated, leaving further explorations as future work. These assumptions simplify the CPO algorithm as follows:

**Proposition 4.1.** *Assuming that $\pi_0(c|x)$ follows a uniform distribution over binary concepts, we have:*

$$\mathcal{L}_{CPO} \propto -\mathbb{E}_{c,x\sim D, c'\neq c\sim\pi_\theta} \left[ \log(\pi_\theta(c|x)) \right].$$
(6)

A proof for this proposition is given in App. C.1. Simply put, the above states that $\mathcal{L}_{\text{CPO}}$ is proportional to optimizing the binary cross-entropy when $\pi_\theta$ samples concepts that differ from those in the empirical distribution. $\mathcal{L}_{\text{CPO}}$ is proportional to the objective in Equation 6, and not equal, because when the sampled concepts are equal to the empirical ones, the objective is constant, i.e., $\log \frac{\pi_\theta(c|x)}{\pi_0(c|x)} - \log \frac{\pi_\theta(c'|x)}{\pi_0(c'|x)} = 0$ in Equation 5. The equivalence to the log-likelihood when

---

[2]In practice, we use hard Gumbel-Softmax sampling (Jang et al., 2017) to ensure end-to-end differentiability. Here, we sample a *single* concept for each image in each iteration, but one could potentially sample multiple per image to increase performance.

the sampled concepts are not equal to the empirical ones relies on assuming a uniform prior.

**Gradient Analysis** Proposition 4.1 highlights a similarity between $\mathcal{L}_{\text{CPO}}$ and $\mathcal{L}_{\text{BCE}}$. Therefore, we can study their gradients to understand their key differences better. Under our previous assumptions, we can express the expected gradient of $\mathcal{L}_{\text{CPO}}$ as

$$\mathbb{E}[\nabla_\theta \mathcal{L}_{\text{CPO}}] = \frac{1}{N} \sum_{\substack{(c,x)\sim\mu \\ c'\sim\pi_\theta}} (\pi_\theta(c|x) - 1)\pi_\theta(c'|x)\nabla_\theta k_\theta$$

$$= \frac{1}{N} \sum_{(c,x)\sim\mu} (\pi_\theta(c|x) - 1)\left(1 - \pi_\theta(c|x)\right)\nabla_\theta k_\theta$$

where $k_\theta$ refers to the pre-trained image encoder traditionally used to generate concept representations. A full derivation of this equality, which exploits the fact that we only have a nonzero gradient when $c' \neq c$ and thus $\pi(c'|x) = 1 - \pi(c|x)$, is given in App. C.1.

This result shows that $\mathcal{L}_{\text{CPO}}$'s gradient is $\mathcal{L}_{\text{BCE}}$'s gradient weighted by how confident the policy is in the sampled concept. This yields the following bound on the CPO gradient:

**Proposition 4.2.** *Under the same assumption as Proposition 4.1, the expected norm of the $\mathcal{L}_{CPO}$'s gradient is a lower bound of $\mathcal{L}_{BCE}$'s expected gradient. That is:*

$$\left\| \mathbb{E}[\nabla_\theta \mathcal{L}_{CPO}] \right\|_2 \leq \left\| \mathbb{E}[\nabla_\theta \mathcal{L}_{BCE}] \right\|_2$$

*Proof.* As $0 \leq \pi(c|x) \leq 1$, we have that

$$\left\| \sum_{(c,x)\sim\mu} (\pi(c|x) - 1)(1 - \pi(c|x)) \right\|_2 \leq \left\| \sum_{(c,x)\sim\mu} (\pi(c|x) - 1) \right\|_2$$

Notice how the right-hand side is equivalent to $\nabla_\theta \mathcal{L}_{\text{BCE}}$ with equality only holding when $c_i \neq c_i'$ for all $i$. Thus, we can see that an implication of not assuming the correctness of the concepts is that $\mathcal{L}_{\text{CPO}}$ is more conservative in its gradient updates than $\mathcal{L}_{\text{BCE}}$. This means that $\mathcal{L}_{\text{CPO}}$ has a larger gradient when $\pi_\theta$ is confident in the sampled concepts and is more conservative when it is uncertain. A visualization of the differences in the gradients is given in App. D. Next, we discuss the direct implications of these results and the relationship to the improved label noise robustness.

## 4.2. Noisy Concept Labels

To improve performance, CBMs are traditionally trained by randomly augmenting input images with transformations, such as cropping or blurring, which may obscure the represented concept. As a result, CBMs are often trained with some level of concept noise, regardless of the reliability of the empirical data. Moreover, commonly used benchmark datasets for CBMs, such as CUB (Wah et al., 2011) and AwA2 (Xian et al., 2019), are designed so that their images

(a) Ours      (b) Kim et al. (2024) Vandenhirtz et al. (2024)

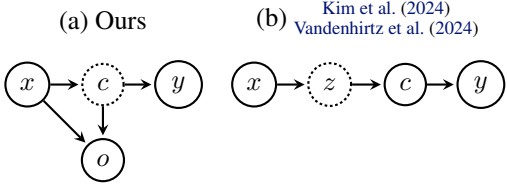

*Figure 2.* Comparison of graphical models for Bayesian CBMs. Dashed outlines indicate the variable over which an amortized posterior is taken. On the left (a), it can be seen that optimizing a CBM using $\mathcal{L}_{CPO}$ directly approximates the posterior distribution of the concepts. On the right (b), it can be seen how other methods instead obtain the posterior over a latent variable $z$.

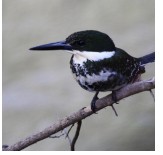 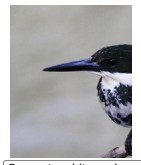 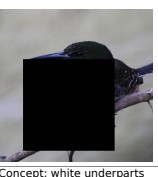

Concept: white underparts

Uncertainty Scores (σ):
  BCE: 0.00 (c-val: 1.00)
  CPO: 0.82 (c-val: 0.71)
  ProbCBM: 0.68 (c-val: 1.00)

Concept: white underparts

Uncertainty Scores (σ):
  BCE: 0.00 (c-val: 1.00)
  CPO: 0.53 (c-val: 0.84)
  ProbCBM: 0.76 (c-val: 1.00)

Concept: white underparts

Uncertainty Scores (σ):
  BCE: 0.04 (c-val: 0.01)
  CPO: 1.00 (c-val: 0.53)
  ProbCBM: 0.67 (c-val: 1.00)

*Figure 3.* Example of uncertainty estimates for different models analyzing the bird's white underbelly. We measure uncertainty ($\sigma$) and model predictions ($c$-val) before and after two augmentations—cropping and blocking—across three models: a CBM trained with $\mathcal{L}_{CPO}$, one with $\mathcal{L}_{BCE}$, and a ProbCBM. The BCE model is consistently confident, even after both augmentations. The ProbCBM's uncertainty stays fairly constant, increasing slightly when zoomed in. In contrast, the $\mathcal{L}_{CPO}$ model becomes more certain with cropping and most uncertain when the concept is blocked.

may not accurately reflect their concept labels. Intuitively, $\mathcal{L}_{CPO}$ dropping the assumption of correctness towards preferences should help under both noisy and optimal conditions. To illustrate why this is the case, we analyze the gradients of $\mathcal{L}_{CPO}$ and $\mathcal{L}_{BCE}$ in the presence of noise.

**Empirical Best Gradient**  To show why $\mathcal{L}_{CPO}$ is more resilient to noise, we examine both losses' gradients under noisy conditions and compare them to their optimal counterparts. To do so, we first make the assumption that $d^*(c, x)$ is one if $x$ and $c$ are the ground truth concepts in the image and zero otherwise. Then, given the empirical and optimal distributions $\mu$ and $d^*$, respectively, we can derive the expected gradient that approximates the ground truth as:

$$\mathbb{E}_{(c^*,x)\sim d^*}[\nabla_\theta \mathcal{L}] = \mathbb{E}_{(c,x)\sim \mu}\left[\frac{d^*(c,x)}{\mu(c,x)}\nabla_\theta \mathcal{L}(c, \pi_\theta(c|x))\right]$$
$$= \mathbb{E}_{(c^*,x)\sim \mu^+}\left[\nabla_\theta \mathcal{L}(c^*, \pi_\theta(c|x))\right].$$

Here, $\frac{d^*(c,x)}{\mu(c,x)}$ is an importance sampling coefficient that equals 1 when $c$ exists in both $d^*$ and $\mu$, and 0 otherwise, as we assume both $\mu$ and $d^*$ are deterministic, and $\mu^+ \in \mu$ is the subset containing only optimal concepts. Conversely, $\mu^- \in \mu$ is the subset containing only suboptimal concepts $c^-$. The resulting gradient on the empirical data is

$$\mathbb{E}_{(c,x)\sim \mu}[\nabla_\theta \mathcal{L}] = \mathbb{E}_{(c^*,x)\sim \mu^+}\left[\nabla_\theta \mathcal{L}(c^*, \pi_\theta(c^*|x)\right]$$
$$+ \mathbb{E}_{(c^-,x)\sim \mu^-}\left[\nabla_\theta \mathcal{L}(c^-, \pi_\theta(c^-|x)\right].$$

It is a linear combination of the gradients on the noisy concepts $c^-$ and the optimal ones $c^*$. This formulation allows us to analyze the difference between the optimal gradient and the gradient produced on a noisy distribution for both $\mathcal{L}_{CPO}$ and $\mathcal{L}_{BCE}$:

**Theorem 4.3.** *The gradient of $\mathcal{L}_{CPO}$ under a constant level of noise is closer in distance to its noise-free counterpart than the gradient of $\mathcal{L}_{BCE}$ under the same noise is to its*

*respective noise-free counterpart. In other words:*

$$\left\|\mathbb{E}_{(c^*,x)\sim d}[\nabla_\theta \mathcal{L}_{CPO}] - \mathbb{E}_{(c,x)\sim \mu}[\nabla_\theta \mathcal{L}_{CPO}]\right\|_2$$
$$\leq \left\|\mathbb{E}_{(c^*,x)\sim d}[\nabla_\theta \mathcal{L}_{BCE}] - \mathbb{E}_{(c,x)\sim \mu}[\nabla_\theta \mathcal{L}_{BCE}]\right\|_2$$

We prove this theorem in App. C.2. Intuitively, Theorem 4.3 says that when examining the difference between optimal and noisy gradients, only terms from noisy observations remain. Thus, according to Proposition C.1, $\left\|\mathbb{E}_{c^-\sim \mu^-}[\nabla_\theta \mathcal{L}_{DPO}]\right\|_2 \leq \left\|\mathbb{E}_{c^-\sim \mu^-}[\nabla_\theta \mathcal{L}_{BCE}]\right\|_2$. This implies that $\mathcal{L}_{CPO}$'s gradient updates more closely approximate their optimal gradients, resulting in better noise robustness.

A simpler explanation lies in the update mechanisms, where CPO only modifies the policy when concepts are incorrectly sampled, creating situations where sampled concepts $c'$ align with $c^-$ and, thus, minimizing noise impact. In contrast, BCE updates continuously unless $\pi_\theta(c|x)$ exactly equals 1 or 0, making it inherently more susceptible to noise. Figure 1 illustrates these results.

### 4.3. Relationship to Amortized Posterior Approximation

Given their relationship, we seek to understand the fundamental difference between optimizing $\mathcal{L}_{CPO}$ and $\mathcal{L}_{BCE}$.

**Control as Inference**  The bottleneck nature of CBMs is similar to that of a Variational Auto-Encoder (VAE) (Kingma & Welling, 2022). Traditionally, such Bayesian methods introduce a "bottleneck" in their inference that is formed by latent variables that are learned in an unsupervised fashion (Doersch, 2016). The graphical model representing a CBM often resembles this relationship, with the key difference that CBMs directly specify the factors within the bottleneck in the form of known concepts. One important outcome of this difference is that CBMs

*Table 1.* Task and concept performances. The highest and second-highest values in each column are bolded and underlined, respectively. We find that $\mathcal{L}_{\text{CPO}}$ consistently achieves improved task accuracy and concept AUC.

| | CUB | | AwA2 | | CelebA | |
|---|---|---|---|---|---|---|
| | **Task Accuracy** | **Concept AUC** | **Task Accuracy** | **Concept AUC** | **Task Accuracy** | **Concept AUC** |
| ProbCBM Sequential | $0.742 \pm 0.004$ | $0.900 \pm 0.007$ | $0.891 \pm 0.003$ | $0.960 \pm 0.003$ | $0.302 \pm 0.008$ | $\mathbf{0.878 \pm 0.006}$ |
| ProbCBM Joint | $0.766 \pm 0.012$ | $0.943 \pm 0.006$ | $0.860 \pm 0.017$ | $0.945 \pm 0.007$ | $0.288 \pm 0.023$ | $0.863 \pm 0.005$ |
| CoopCBM | $0.760 \pm 0.004$ | $0.936 \pm 0.001$ | $0.888 \pm 0.006$ | $0.950 \pm 0.003$ | $0.288 \pm 0.011$ | $0.878 \pm 0.002$ |
| CBM BCE | $0.753 \pm 0.009$ | $0.937 \pm 0.001$ | $0.900 \pm 0.008$ | $0.959 \pm 0.003$ | $0.283 \pm 0.007$ | $0.873 \pm 0.002$ |
| CBM CPO (Ours) | $\underline{0.800 \pm 0.003}$ | $\mathbf{0.952 \pm 0.001}$ | $\underline{0.915 \pm 0.004}$ | $\mathbf{0.971 \pm 0.001}$ | $0.310 \pm 0.009$ | $0.857 \pm 0.003$ |
| CEM BCE | $\underline{0.800 \pm 0.003}$ | $\underline{0.946 \pm 0.001}$ | $0.889 \pm 0.001$ | $0.953 \pm 0.000$ | $\underline{0.351 \pm 0.006}$ | $\underline{0.875 \pm 0.004}$ |
| CEM CPO (Ours) | $\mathbf{0.807 \pm 0.004}$ | $0.931 \pm 0.003$ | $\mathbf{0.917 \pm 0.003}$ | $\underline{0.965 \pm 0.001}$ | $\mathbf{0.352 \pm 0.004}$ | $0.853 \pm 0.003$ |

are traditionally trained to optimize the likelihood of the empirical concepts, which is *fundamentally different* from approximating the concept's posterior distribution (Koller & Friedman, 2009). On the other hand, Haarnoja et al. (2017) show that Equation 1 — and Equation 5 by extension — is equivalent to training an amortized posterior approximation of the actions (concepts in our contexts). This derivation relies on introducing an optimality latent variable $o$ whose relationship to $x$ and $c$ is visualized in Figure 2 (Eysenbach & Levine, 2022; Levine, 2018). This optimality variable denotes whether or not the given state-action pair sampled from $\pi$ is optimal $o = 1$ ($c$ is the best visually represented concept in $x$) or not $o = 0$. The distribution over this variable is then given as:

$$p(o = 1|x, c) = \exp(r^*(c, x)) \tag{7}$$

where $r^*(c, x) \in (-\infty, 0]$ in our case is an unknown reward function indicating how well a given concept set is represented in an image. Here, one can interpret $p(o = 1|x, c)$ as the probability that the given concept set $c$ is correct, or *optimal*, for input $x$, and $p(o = 1|x)$ as how optimal, on average, the concept sets sampled from $\pi$ are for a given $x$. Given this, a posterior over the concepts is:

$$\pi(c|o = 1, x) = \frac{p(o = 1|c, x)\pi_0(c|x)}{p(o = 1|x)} \tag{8}$$

$$= \frac{1}{Z(o)}\pi_0(c|x) \exp\left(\frac{1}{\beta}r^*(x, c)\right) \tag{9}$$

where $Z(o) = p(o = 1|x)$. This objective is equivalent to that in Equation 2 as $Z(o)$ must be equivalent to $Z(x)$ for $\pi(c|o = 1, x)$ to be a valid probability distribution. Hence, optimizing $\pi_\theta$ using the objective in Equation 1 - and Equation 5 by extension - *directly approximates the optimal concept posterior distribution* where $\pi^*(c|x) = \pi(c|o = 1, x)$.

We provide a comprehensive analysis of uncertainty quantification in App. G, where we evaluate the uncertainty performance of $\mathcal{L}_{\text{CPO}}$ against baselines. Overall, both quantitative and qualitative results show that $\mathcal{L}_{\text{CPO}}$ offers better

uncertainty estimates than other models, being more sensitive to obstructions of the target object/concept. Figure 3 presents a brief case study on an example image, comparing a vanilla CBM trained with $\mathcal{L}_{\text{CPO}}$, one trained with $\mathcal{L}_{\text{BCE}}$, and a ProbCBM (Kim et al., 2023), which amortizes the posterior over a hidden variable (see the right side of Figure 2). We analyze the uncertainty scores $\sigma$—the normalized variance scores for each model (variance of a Bernuilli variable for CBMs and determinant of covariance for ProbCBM)—and the $c$-val, representing the predictions of $\pi_\theta(c|x)$. This example illustrates a common trend from our quantitative analysis: $\mathcal{L}_{\text{CPO}}$ more effectively increases its uncertainty when the target object (in this case, the bird) is obstructed, compared to other baselines.

## 5. Experiments

Here, we validate the $\mathcal{L}_{\text{CPO}}$ objective in three different settings. First, we study $\mathcal{L}_{\text{CPO}}$ in clean, optimal data, then under concept label noise, and finally in a streaming data context where we leverage a prior when computing our updates.

**Datasets** We study our proposed objective on three real-world image datasets: Caltech-UCSD Birds-200-2011 (CUB) (Wah et al., 2011), Large-scale CelebFaces Attributes (CelebA) (Liu et al., 2015), and Animals with Attributes 2 (AwA2) (Xian et al., 2019). For CUB, we use the 200 class labels and 112 concept labels used by (Koh et al., 2020). For AwA2, we use the original 50 classes and each sample's 85 attributes as concept labels. Finally, for CelebA, we use the 256 classes and six concepts selected by Espinosa Zarlenga et al. (2022). The latter dataset is included to study our approach in a setting where the concept set is not fully descriptive (i.e., *complete*) of the downstream task. Further details of each dataset can be found in App. B.

**Baselines** We evaluate $\mathcal{L}_{\text{CPO}}$ against $\mathcal{L}_{\text{BCE}}$ on the following CBM-based architectures: (1) standard joint CBMs with sigmoidal concept representations (CBM), Concept Embedding Models (CEMs) (Espinosa Zarlenga et al.,

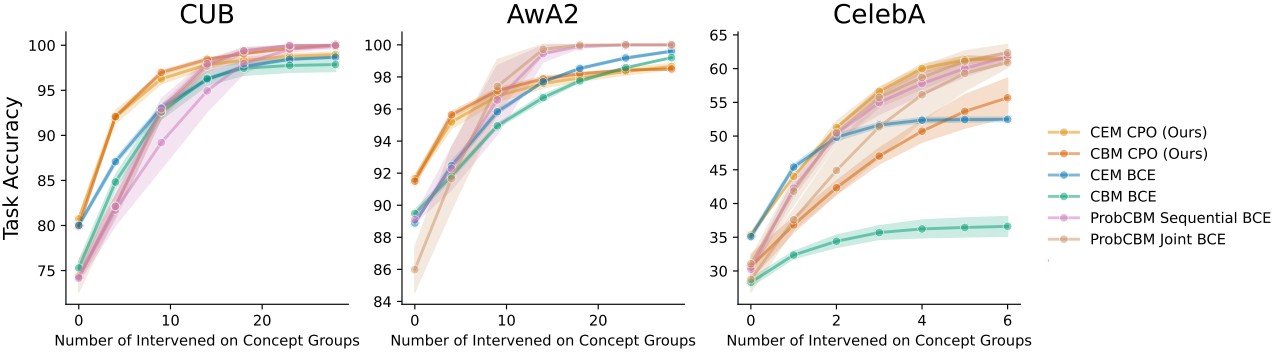

*Figure 4.* CUB Interventions without added label noise. In CUB and CelebA, $\mathcal{L}_{\text{CPO}}$ models lead to the best intervention performance. While in AwA2, a substantial ($\sim$8-15) number of interventions must be performed for $\mathcal{L}_{\text{BCE}}$-based models to outperform $\mathcal{L}_{\text{CPO}}$ ones.

2022), which employ more expressive concept representations to increase model capacity, (3) Coop-CBMs (Sheth & Ebrahimi Kahou, 2024), which use an auxiliary head to improve a CBM's information bottleneck, and (4) ProbCBMs (Kim et al., 2023), which introduce a latent parameter between inputs and concepts (Figure 2). ProbCBMs, in particular, allow us to compare amortizing the concepts' posterior distribution instead of a latent variable's. However, while ProbCBMs traditionally use sequential training, *we jointly train them* to ensure a fair comparison across baselines. We do not evaluate training traditional ProbCBMs with $\mathcal{L}_{\text{CPO}}$ as their loss function optimizes an evidence-lower-bound on the latent variables' posterior, which explicitly requires maximizing concept likelihood. We note that while $\mathcal{L}_{\text{CPO}}$ introduces a new parameter $\beta$, we choose not to tune it and set $\beta = 1$ for all experiments. Overall, this means we tune the same number of hyper-parameters for CBMs trained with $\mathcal{L}_{\text{CPO}}$ and $\mathcal{L}_{\text{BCE}}$. We discuss other hyperparameters and implementation details in App. A.

### 5.1. Un-noised Evaluation

We first evaluate the task accuracy and mean concept AUC-ROC of models trained with $\mathcal{L}_{\text{CPO}}$ under the traditional CBM setting, where no additional label noise is added. Thereafter, we analyze intervention performance.

**Base Performance** Table 1 summarizes performance metrics for each baseline and dataset. Our results suggest that training with the $\mathcal{L}_{\text{CPO}}$ objective enhances the base task accuracy of both standard CBMs and CEMs with minimal-to-no-loss in mean concept AUC. Most notably, we observe that, in CUB and AwA2, CPO-trained CBMs match/outperform CEMs trained with BCE, a significant result since these benefits from CPO come *without any* additional parameters or significant computational overheads (see App. I). Finally, we find that Coop CBM performs similarly to traditional CBMs, finding no significant difference between them. Due

to this and similar findings in preliminary results, we do not evaluate this method in the remaining experiments and further discuss Coop CBM's performance in App. B.1.

**Interventions** A key advantage of CBMs is their ability to improve their task performance through test-time *concept interventions*. In § 4.3 we show that $\mathcal{L}_{\text{CPO}}$ directly optimizes for the concept posterior. Thus, an advantage of this is that we can obtain accurate uncertainty estimates from the predicted concept values. To test the effectiveness of this uncertainty estimate, we study the effect of interventions when we choose the order in which concepts are intervened on based on their uncertainties, i.e., more uncertain concepts are intervened on first. We do this for similar approaches by using the concept prediction as an uncertainty estimate for CBMs and the determinant of the covariance matrix for ProbCBMs (as done by Kim et al. (2023)).

Figure 4 illustrates the responsiveness of models to interventions. Here, we see that, across all datasets, CEMs and standard CBMs trained with $\mathcal{L}_{\text{CPO}}$ exhibit better accuracies as they are intervened on than their $\mathcal{L}_{\text{BCE}}$ counterparts. This suggests that directly modeling the concept posterior distribution provides better uncertainty estimates, leading to more effective interventions. Additionally, CBMs and CEMs trained with $\mathcal{L}_{\text{CPO}}$ achieve stronger intervention performance than ProbCBMs on CUB, while CEMs using $\mathcal{L}_{\text{CPO}}$ outperform ProbCBMs on CelebA. The only exception is AwA2, where ProbCBMs, on average, still require approximately eight interventions before surpassing $\mathcal{L}_{\text{CPO}}$ models.

### 5.2. Noised Evaluation

Next, we empirically study $\mathcal{L}_{\text{CPO}}$ under various amounts of noise. To do this, we randomly flip each training concept label with probability $p$ and study the resulting models.

**Base Performance Under Noise** Figure 5 shows our baselines' task accuracies and concept AUCs as we ablate the

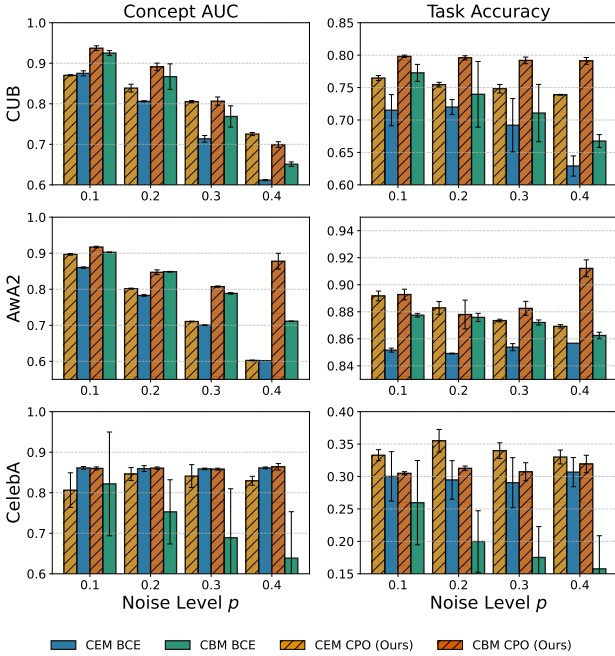

*Figure 5.* Performance metrics for all tasks across varying degrees of label noise. We find that across all noise levels, $\mathcal{L}_{\text{CPO}}$ consistently alleviates performance loss.

label noise probability $p$ across $\{0.1, 0.2, 0.3, 0.4\}$. We observe that, under noisy label conditions, training CBMs and ProbCBMs (see App. F.2) with $\mathcal{L}_{\text{BCE}}$ leads to a significant drop in task accuracy and concept AUC (except for AwA2). Interestingly, we see that CEMs trained with $\mathcal{L}_{\text{BCE}}$ are much more resilient to noise in comparison to CBMs. However, we still observe significant drops in concept AUCs in CEMs trained with $\mathcal{L}_{\text{BCE}}$ in all tasks but CelebA. We believe CEMs' more robust performance in CelebA is due to the bottleneck imposed by this dataset being small (only 6 concepts), making any additional capacity extremely helpful. In contrast, we see that models trained with $\mathcal{L}_{\text{CPO}}$ are very resilient to noise. Specifically, we find that in terms of task accuracy, CBMs trained with $\mathcal{L}_{\text{CPO}}$ are the least affected by noise and largely surpass the performance of CEMs. Moreover, we find that $\mathcal{L}_{\text{CPO}}$-trained models have their concept AUCs better preserved, consistently holding the best or second-best ranks in concept AUC, often attaining significantly better concept AUCs than $\mathcal{L}_{\text{BCE}}$-based models. Most interestingly, we find that even at rather noisy levels ($p = 0.4$), CBMs trained with $\mathcal{L}_{\text{CPO}}$ can outperform more complex models trained with $\mathcal{L}_{\text{BCE}}$ and, in some cases, are largely unaffected by the noise. Overall, we find that using $\mathcal{L}_{\text{CPO}}$ is an effective way to counteract concept label noise.

**Interventions Under Noise** While we find that in the presence of label noise models trained with $\mathcal{L}_{\text{CPO}}$ achieve better performances, such findings are less meaningful if

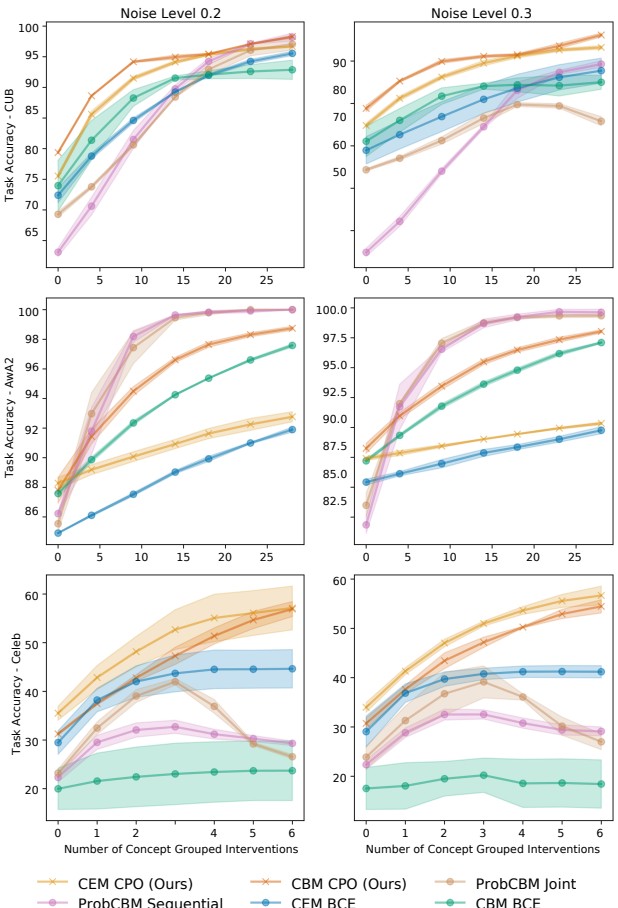

*Figure 6.* Intervention performance under noisy settings. We find that $\mathcal{L}_{\text{CPO}}$ is generally less affected by concept label noise in terms of interventionality. This finding is specifically relevant to CUB and CelebA where we find little to no performance degradation when compared to the un-noised setting.

they come at the cost of a CBM's intervenability. To this end, in Fig. 6 we report the intervention performance as we vary training label noise. We observe that models trained with $\mathcal{L}_{\text{BCE}}$ have their intervenability significantly affected by noise, with CEM the occasional exception (particularly in CelebA). In App. F.1 we visualize the intervention performance for all $p \in \{0.1, 0.2, 0.3, 0.4\}$ finding that in stark contrast to $\mathcal{L}_{\text{BCE}}$, models trained with $\mathcal{L}_{\text{CPO}}$ remain intervenable under the presence of noise, only being to severely impacted at high noise rates (e.g., $p = 0.4$).

### 5.3. Noise by Labeler Confidence

While randomly noising the concept labels provides allows us to analyze how models may perform under noise without any prior on the structure of the noise, this may sometimes not be the most realistic scenario as noise is often structured. For this we here we analyze noising based on the confidence

score of the labeler provided for each concept in each image in the CUB dataset. This allows us to apply noise based on perceived uncertainty. The dataset provides a confidence rating from 1 to 4, where 1 is the least confident and 4 is the most. We apply noise proportionally to this score: a rating of 4 corresponds to $p = 0.1$, 3 to $p = 0.2$, and so on. Additionally, in App. H another structured noising procedures, where we noise concepts at the concept group level.

Table 2 shows the results when concepts are noised according to their confidence levels. Interestingly, much like in the randomly noised setting, we find that $\mathcal{L}_{\text{CPO}}$ again outperforms the $\mathcal{L}_{\text{BCE}}$ counterparts in both task accuracy and concept AUC greatly reducing the impact of noise.

| Model | Task Accuracy | Concept AUC |
|---|---|---|
| CBM BCE | $0.733 \pm 0.032$ | $0.876 \pm 0.012$ |
| CBM CPO | $\mathbf{0.793} \pm 0.002$ | $\mathbf{0.913} \pm 0.000$ |
| CEM BCE | $0.704 \pm 0.053$ | $0.831 \pm 0.048$ |
| CEM CPO | $\underline{0.757} \pm 0.004$ | $\underline{0.846} \pm 0.006$ |

*Table 2.* Noising proportionally to the confidence level of the labeler (e.g certainty of 4 equals 0.1 noise 3, 0.2 and so forth). We find that $\mathcal{L}_{\text{CPO}}$ is substantially able to alleviate the effects of this noise specifically on task accuracy.

### 5.4. Learning on Streaming Data

A byproduct of $\mathcal{L}_{\text{CPO}}$ optimizing for an approximate posterior is its ability to leverage a prior. So far, we have assumed a uniform prior over concepts, but here we explore adjusting it. One key benefit of CBMs is that practitioners can scrutinize concept representations at test time, enhancing trust, accuracy, and enabling the ability to collect new training data through interventions. Specifically, when a CBM is intervened on, it obtains a new concept label that can be used to improve the system further (an idea that has been explored in other fields (Stephan et al., 2024; Shi et al., 2024)). To explore this, we first partition the training data of CUB into four evenly sized blocks, of which we use the first block (25% of the data) to train a joint CBM using $\mathcal{L}_{\text{CPO}}$ on the task labels $y$ and the concepts $c$. Thereafter, we analyze training on the remaining data blocks *only* using concept labels in three different ways: using $\mathcal{L}_{\text{BCE}}$, $\mathcal{L}_{\text{CPO}}$ with a uniform prior and $\mathcal{L}_{\text{CPO}}$ with the previous checkpoint as the prior. The main idea is that a prior can help prevent the model from drifting too far from the policy jointly trained with the task predictor $f_\phi$.

Figure 7 evaluates models using $k\% \in \{50\%, 100\%\}$ of total concepts. Curiously, we find that $\mathcal{L}_{\text{CPO}}$ using a uniform prior performs worse when using more concept labels. We believe this is due to the policy drifting further from that of the initial checkpoint. We find that indeed by using a prior

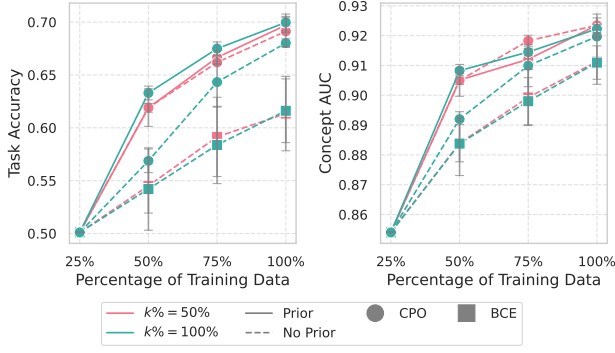

*Figure 7.* Updating a CBM with streaming concept labels (no task labels). We compare $\mathcal{L}_{\text{BCE}}$, $\mathcal{L}_{\text{CPO}}$ (No Prior), and $\mathcal{L}_{\text{CPO}}$ (Prior) using a $\mathcal{L}_{\text{CPO}}$-trained initial policy with $k\% \in \{50\%, 100\%\}$ of concept groups. We find updating $\mathcal{L}_{\text{CPO}}$ with a prior leverages all concept labels in this setting.

one can alleviate this drift and enable $\mathcal{L}_{\text{CPO}}$ to leverage all new concept labels. While here we find $\mathcal{L}_{\text{BCE}}$ underperforms $\mathcal{L}_{\text{CPO}}$, in App. 7 we show this gap narrows—though not fully closes—when the initial policy is trained with $\mathcal{L}_{\text{BCE}}$.

## 6. Discussion

**Future Work** In this work, we aim to maintain end-to-end differentiability by leveraging a DPO-like objective, but this requirement is not strictly necessary. Differentiability is mainly needed due to the absence of a reward function; however, one could instead employ variable rewards (e.g., based on visual consistency) or learn a reward function from preference pairs (as done in language modeling). Overall, we believe RL techniques offer a promising direction for CBM training—potentially enhancing robustness and enabling diverse behaviors through non-differentiable objectives.

**Conclusion** We present a DPO-inspired training objective for CBMs called $\mathcal{L}_{\text{CPO}}$. Our loss directly optimizes for the concept's posterior distribution, with concept representations that explicitly encode uncertainty, leading to improved intervention performance. We provide analysis demonstrating that $\mathcal{L}_{\text{CPO}}$ exhibits greater robustness to noise compared to $\mathcal{L}_{\text{BCE}}$ and empirically show that a simple CBM trained with the $\mathcal{L}_{\text{CPO}}$ objective can consistently outperform competing methods without any additional parameters. Moreover, our experiments complement our analysis on $\mathcal{L}_{\text{CPO}}$'s behaviour under noise by showing that $\mathcal{L}_{\text{CPO}}$ yields better concept AUC and task accuracy than BCE-based models while maintaining its intervention performance. Furthermore, we demonstrate how the $\mathcal{L}_{\text{CPO}}$ objective's prior can be leveraged to learn more efficiently from streaming data. Ultimately, $\mathcal{L}_{\text{CPO}}$ offers numerous benefits for CBM and CBM-like methods with minimal computational overhead.

## Impact Statement

Concept bottleneck models (CBMs) have emerged as a promising approach to increasing the trustworthiness and transparency of AI systems, areas where modern machine learning methods often fall short. This work takes a step toward improving the robustness of such models. While CBMs enhance the interpretability of AI systems, we demonstrate that they are highly sensitive to concept mislabeling—a significant limitation in high-impact domains like healthcare and law enforcement, where labels are often noisy and inherently subjective. We show that our proposed objective effectively addresses this issue, enabling CBMs to perform more reliably in real-world applications and expanding their potential for meaningful impact.

## Acknowledgments

We thank the several anonymous reviewers for their feedback which has greatly helped improve this work. EP acknowledges the support of the NSERC PGS-D grant and the Bourse en intelligence artificielle provided by Université de Montréal. TZ acknowledges the support of Samsung Electronics Co., Ldt. MEZ acknowledges support from the Gates Cambridge Trust via a Gates Cambridge Scholarship. LC recognizes the support of the Canada CIFAR AI Chair Program, the Canada First Research Excellence Fund and IVADO. We also acknowledge that this research was enabled in part by computing resources, software, and technical assistance provided by Mila.

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

# A. Implementation Details

## A.1. Tuning

We employ a ResNet34 (He et al., 2015) as the backbone image encoder $k_\theta$, pretrained on ImageNet-1k (Russakovsky et al., 2015). Following standard procedures, we apply random cropping and flipping to a portion of the images during training. This augmentation process may introduce non-zero noise levels, as some concepts could be removed from the images after transformation. We use a batch size of 512 for the Celeb dataset and 256 for for CUB and AwA2. We train all models using RTX8000 Nvidia-GPU. In all datasets we train for up to 200 epochs and early stop if the validation loss has not improved in 15 epochs. For fair evaluation across methods, we tune the learning rate for CEMs, CBMs, and ProbCBM. Specifically, for CUB and AwA2 datasets, we explore learning rates $\in \{0.1, 0.01\}$, while for CelebA, we expand the search to $\in \{0.1, 0.01, 0.05, 0.005\}$ due to the observed instability of CEMs at higher learning rates. Additionally, we set the hyper-parameter $\lambda \in \{1, 5, 10\}$ for all methods. For CEMs and models trained using $\mathcal{L}_{\text{DPO}}$, we found RandInt beneficial, which randomly intervenes on $25\%$ of the concepts during training. ProbCBM introduce a few extra hyperparameters which in this work we did not tune and directly use the hyper-parameters provided by the original authors. Similar to other models, ProbCBM employs RandInt at $50\%$, making it particularly sensitive to interventions, especially in concept-complete tasks such as AwA2 and CUB. The only model for which we tune additional hyper-parameters is Coop-CBM, where we adjust the weight parameter for the auxiliary loss we discuss more in detail in App B.1. All experiments are ran using a forked version of the Github[3] repository used by Espinosa Zarlenga et al. (2022).

# B. Datasets

**CUB (Wah et al., 2011).** In CUB we use the standard dataset used in (Koh et al., 2020) madeup of $k = 112$ concept annotations representing bird attributes (e.g., beak type, wing color) and use the bird class ($m = 200$) as the downstream task. Our only departure from Koh et al. (2020) is that we group the concepts into 28 semantic concept groups, following Espinosa Zarlenga et al. (2022). We use the same image processing as in (Koh et al., 2020) and by randomly flipping and cropping some images during training. The final dataset is composed of $\sim 6,000$ RGB images of dimension $(3, 299, 299)$, and split into a standard 70%-10%-20% train-validation-test split.

**AwA2 (Xian et al., 2019).** For AwA2 we use the same data processing as Xu et al. (2024). Which are made up of where the $k = 85$ concepts correspond to visual animal attributes (e.g., has wings, has claws) which are grouped into 28 semantic concept groups. We apply standard rotation and cropping augmentations throughout training and use the standard 70%-10%-20% train-validation-test split.

**CelebA (Liu et al., 2015).** For this dataset, we closely follow the data processing done by (Espinosa Zarlenga et al., 2022), where they select the 8 most balanced attributes out a total of 40 binary attributes. Where they generate $m = 256$ classes by assign them a value based on the base-10 representation of their attribute label. We construct the incomplete concept set using the same 6 attributes selected by (Espinosa Zarlenga et al., 2022). We follow the same subsampling procedure as (Espinosa Zarlenga et al., 2022) and randomly select $\frac{1}{12}$th of the images for training. This results in a final dataset composed of 16,900 RGB images where we use the same 70%-10%-20% train-validation-test split.

## B.1. Coop CBM

Here, we briefly outline the training procedure for Coop-CBM which we found to perform similarly to CBMs trained with $\mathcal{L}_{\text{BCE}}$. Similar to other methods, we tune the learning rate and the concept loss weight $\lambda$. However, Coop-CBM is the only model for which we conduct more extensive hyper-parameter tuning, as we observed minimal differences between it and a standard CBM trained with $\mathcal{L}_{\text{BCE}}$. Specifically, we tune the additional hyper-parameter $\gamma \in \{0.01, 1, 5, 10\}$, which controls the strength of the auxiliary head[4]. In our setup, the optimal values for $\gamma$ were found to be $\gamma = 5$ for CUB, $\gamma = 10$ for AwA2, and $\gamma = 0.01$ for CelebA. We observed negligible differences between Coop-CBM and standard CBMs in terms of base and intervention performance (see Figure 8), except for in AwA2 where it improves intervention performance but outperforms ProbCBM at higher number of interventions. As a result, we did not include Coop-CBM in the remaining experiments.

---

[3]https://github.com/mateoespinosa/cem
[4]Referred to as $\beta$ in their work, but we change the notation to avoid confusion with our $\beta$ parameter.

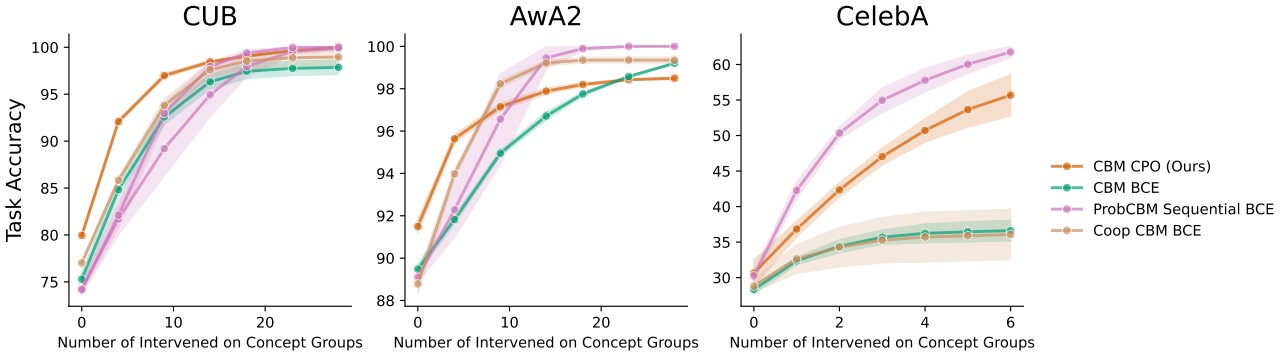

*Figure 8.* Intervention performance including Coop CBMs.

# C. Analysis

While prior work has shown that under mild conditions, offline RL performs (equivalent CPO) better than likelihood-based training under noisy labels (Kumar et al., 2022; Rashidinejad et al., 2021), it still does not fully answer the question as to why specifically CPO should perform better in our context.

## C.1. Gradient Derivations

**Assumption:** For all derivations, we assume binary labels and that $\pi_0$ follows a uniform distribution. Additionally to reduce clutter, in all gradient derivations, we drop the $\nabla k_\theta$ as it does not affect the gradients of the loss functions. We note to simplify the proofs we will assume conditional independence of the concepts i.e., $c_i \perp c_j | x \; \forall \; i \neq j$, but this is not strictly necessary. One could derive the same conclusions using an autoregresive decomposition of the joint density of the concepts.

**Derivation of the CPO objective**

$$\mathcal{L}_{\text{CPO}} = -\mathbb{E}_{c,x\sim D,c'\sim\pi_\theta(c|x)}[\log\sigma(\log\pi_\theta(c|x) - \log\pi_\theta(c'|x)] \tag{10}$$

$$= -\mathbb{E}_{c,x\sim D,c'\sim\pi_\theta(c|x)}[\log\sigma(\frac{\pi_\theta(c|x)}{\pi_\theta(c'|x)})] \tag{11}$$

$$= -\mathbb{E}_{c,x\sim D,c'\sim\pi_\theta(c|x)}[\log 1 - \log(1 + \exp(-\log\frac{\pi_\theta(c|x)}{\pi_\theta(c'|x)}))] \tag{12}$$

$$= \mathbb{E}_{c,x\sim D,c'\sim\pi_\theta(c|x)}[\log(1 + \exp(-\log(\frac{\pi_\theta(c|x)}{\pi_\theta(c'|x)})))] \tag{13}$$

$$= \mathbb{E}_{c,x\sim D,c'\sim\pi_\theta(c|x)}[\log(1 + \frac{\pi_\theta(c'|x)}{\pi_\theta(c|x)})] \tag{14}$$

$$\tag{15}$$

Thus in this case, we can see if $c'$ is not equivalent to $c$, this loss reduces to cross entropy.

$$\mathcal{L}_{\text{CPO}} = \mathbb{E}_{c,x\sim D,c'\neq c\sim\pi_\theta}[\log(\frac{\pi_\theta(c|x) + (1 - \pi_\theta(c|x))}{\pi_\theta(c|x)})] \tag{16}$$

$$= \mathbb{E}_{c,x\sim D,c'\neq c\sim\pi_\theta}[\log(\frac{1}{\pi_\theta(c|x)})] \tag{17}$$

$$= -\mathbb{E}_{c,x\sim D,c'\neq c\sim\pi_\theta}[\log(\pi_\theta(c|x))] \tag{18}$$

$$\tag{19}$$

Otherwise, it reduces to a constant:

$$\mathcal{L}_{\text{CPO}} = \mathbb{E}_{c,x\sim D, c'=c\sim\pi_\theta}[\log(\frac{\pi_\theta(c|x) + \pi_\theta(c|x)}{\pi_\theta(c|x)})] \tag{20}$$

$$= -\log(\frac{1}{2}) \tag{21}$$

$$\tag{22}$$

**CPO Gradient Derivation**

$$\nabla_\theta\mathcal{L}_{\text{CPO}} = \mathbb{E}_{c,x\sim D, c'\neq c\sim\pi_\theta}[\nabla_\theta\log(\frac{\pi_\theta(c|x) + (1 - \pi_\theta(c'|x))}{\pi_\theta(c|x)})] \tag{23}$$

$$\tag{24}$$

Due to the gradient being zero when $c = c'$, the expected gradient of the CPO objective simplifies to:

$$\nabla_\theta\mathcal{L}_{\text{CPO}} = \frac{1}{N}\sum_{(c,x)\sim\mu, c'\neq c\sim\pi_\theta(c|x)}(\pi_\theta(c|x) - 1)\pi_\theta(c'|x) \tag{25}$$

$$= \frac{1}{N}\sum_{(c,x)\sim\mu, c'\neq c\sim\pi_\theta(c|x)}(\pi_\theta(c|x) - 1)(1 - \pi_\theta(c|x)) \tag{26}$$

That is, the CPO objective only takes a gradient step for sampled concepts that do not equal the empirical concepts.

## C.2. Bounding the gradients

**Proposition C.1.** *The expected gradient given by $\mathcal{L}_{CPO}$ under binary labels is strictly less than or equal to the gradient of the $\mathcal{L}_{BCE}$.*

*Proof:* This proof relies strictly on the notion that $1 - \pi(c|x) \leq 1$ thus:

$$\frac{1}{N}\Big\|\sum_{(c,x)\sim\mu,}(\pi_\theta(c|x) - 1)(1 - \pi_\theta(c|x))\Big\|_2 \leq \frac{1}{N}\Big\|\sum_{(c,x)\sim\mu}(\pi_\theta(c|x) - 1)\Big\|_2 \tag{27}$$

Observe how the right-hand side is equivalent to the expected cross-entropy loss. The above proposition also takes into account the maximum gradient possible for the $\mathcal{L}_{\text{CPO}}$, which is when $c_i = c_i'$ for all $i$.

**Theorem C.2.** *The expected squared difference between the optimal gradient and one computed under noisy labels for direct preference optimization is less than or equal to that for binary cross entropy.*

*Proof:* The optimal gradient to take under noisy labels is given by:

$$\mathbb{E}_{(c^*,x)\sim d}[\nabla_\theta \mathcal{L}] = \mathbb{E}_{(c,x)\sim\mu}[\frac{d(c,x)}{\mu(c,x)}\nabla_\theta \mathcal{L}(c,\pi_\theta(c|x))] \tag{28}$$

$$= \mathbb{E}_{(c,x)\sim\mu}[\frac{d(c,x)}{\mu(c,x)}\nabla_\theta \mathcal{L}(c,\pi_\theta(c|x))] \tag{29}$$

$$= \mathbb{E}_{(c^*,x)\sim\mu^+}[\nabla_\theta \mathcal{L}(c^*,\pi_\theta(c|x))] \tag{30}$$

$$\tag{31}$$

We observe that when we do not adjust for the importance weight, the gradient under noisy labels is:

$$\mathbb{E}_{(c,x)\sim\mu}[\nabla_\theta \mathcal{L}] = \mathbb{E}_{(c^*,x)\sim\mu^+}[\nabla_\theta \mathcal{L}(c^*,\pi_\theta(c^*|x)) + \mathbb{E}_{(c^-,x)\sim\mu^-}[\nabla_\theta \mathcal{L}(c^-,\pi_\theta(c^-|x))] \tag{32}$$

$$\tag{33}$$

Thus the difference in the expected value of the gradients is:

$$\left\|\mathbb{E}_{(c^*,x)\sim d}[\nabla_\theta \mathcal{L}] - \mathbb{E}_{(c,x)\sim\mu}[\nabla_\theta \mathcal{L}]\right\|_2 = \left\|\mathbb{E}_{(c^-,x)\sim\mu^-}[\nabla_\theta \mathcal{L}(c^-,\pi_\theta(c^-|x))]\right\|_2 \tag{34}$$

Therefore, using Proposition C.1 we can observe that :

$$\left\|\mathbb{E}_{(c^-,x)\sim\mu^-}[\nabla_\theta \mathcal{L}_{\text{CPO}}(c^-,\pi_\theta(c^-|x))]\right\|_2 \leq \left\|\mathbb{E}_{(c^-,x)\sim\mu^-}[\nabla_\theta \mathcal{L}_{\text{BCE}}(c^-,\pi_\theta(c^-|x))]\right\|_2 \tag{35}$$

And thus:

$$\left\|\mathbb{E}_{(c^*,x)\sim d}[\nabla_\theta \mathcal{L}_{\text{CPO}}] - \mathbb{E}_{(c,x)\sim\mu}[\nabla_\theta \mathcal{L}_{\text{CPO}}]\right\|_2 \leq \left\|\mathbb{E}_{(c^*,x)\sim d}[\nabla_\theta \mathcal{L}_{\text{BCE}}] - \mathbb{E}_{(c,x)\sim\mu}[\nabla_\theta \mathcal{L}_{\text{BCE}}]\right\|_2 \tag{36}$$

# D. Gradient Visualizations:

We empirically verify the results posed in Theorem 4.3. For this, we train a standard CBM where the total loss function is $\mathcal{L}_{\text{total}} = \frac{1}{2}(\mathcal{L}_{\text{CPO}} + \mathcal{L}_{\text{BCE}})$ and *do not* optimize the task loss. We train this model over 100 gradient steps and visualize their gradients throughout training. The optimal gradient for each loss $\mathcal{L}^*$ is computed using the empirical concepts and the full loss $\mathcal{L}^-$ is computed over both noisy and non-noisy data points. To minimize the effects of noise on the labeled data and gain a better approximation, we explicitly *do not* augment the data in any way. Figure 9 visualizes these results for $p \in \{0.1, 0.3\}$, which empirically confirms the proposed theoretical results showing how even under low amounts of noise $p = 0.1$ $\mathcal{L}_{\text{CPO}}$ is a better approximation to its optimal gradient when compared to $\mathcal{L}_{\text{BCE}}$. We find that in higher noise settings $p = 0.3$, $\mathcal{L}_{\text{CPO}}^-$ deviates substantially less to $\mathcal{L}_{\text{CPO}}^*$ compared to $\mathcal{L}_{\text{BCE}}^-$ against $\mathcal{L}_{\text{BCE}}^*$. This difference is specifically evident early on in training. We hypothesize providing better gradients early on in training potentially improves the generalization of the model being a possible explanation for the improved performance of $\mathcal{L}_{\text{CPO}}$ under noise seen in the empirical evaluation.

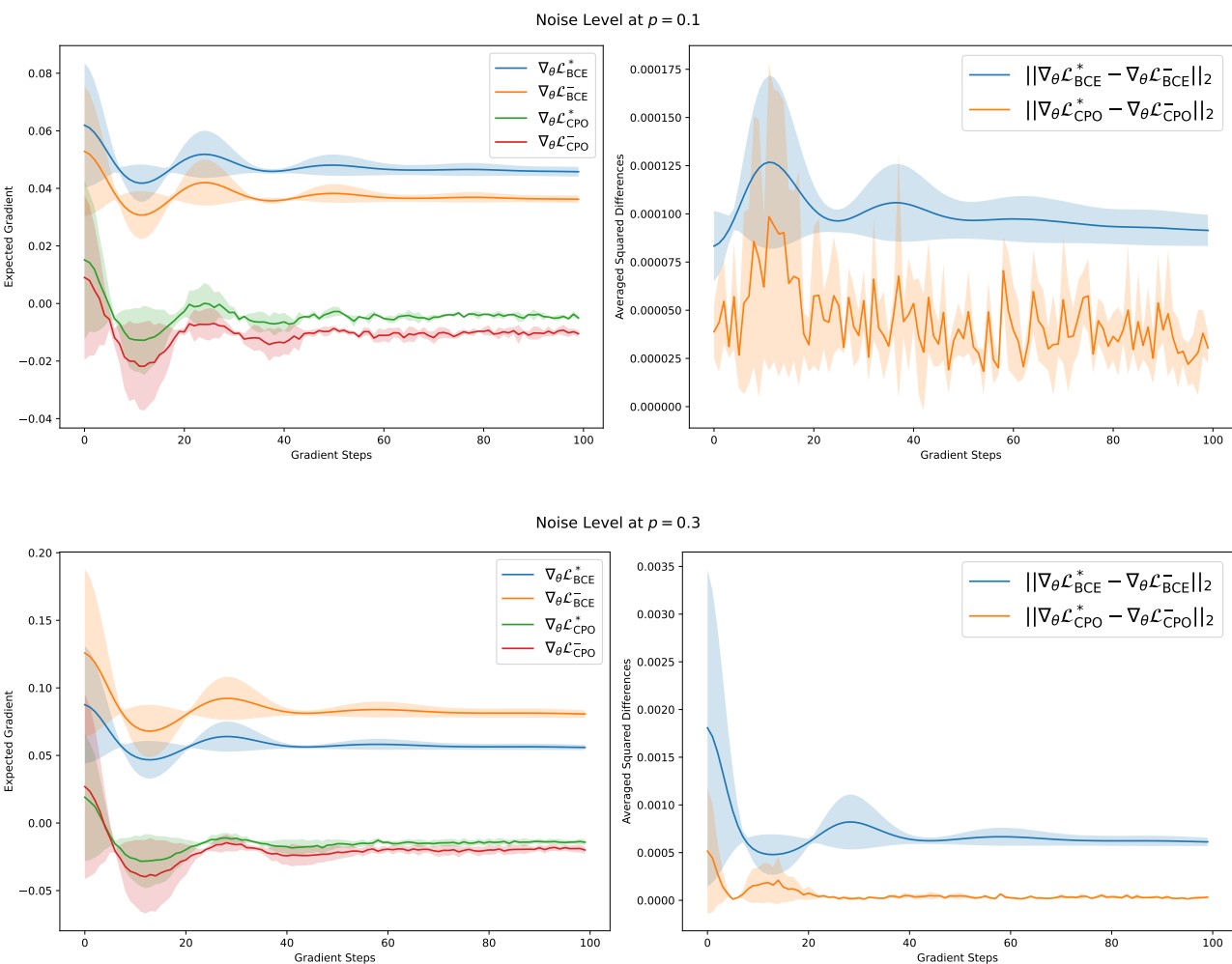

*Figure 9.* Visualization of noisy (indicated by a $^-$) and optimal (indicated with a *) gradients for $\nabla_\theta \mathcal{L}_{\text{CPO}}$ and $\nabla \mathcal{L}_{\text{BCE}}$. We can observe that even in low noise settings $p = 0.1$, $\mathcal{L}_{\text{CPO}}$ better approximates its optimal gradient, with this difference growing as noise level increases especially at the beginning of training. We note that while visually it may seem that the squared difference for $\mathcal{L}_{\text{CPO}}$ is smaller for $p = 0.3$ than that for $p = 0.1$, this is mainly due to scale.

## E. Relationship to Amortized Posterior Approximation

In this appendix, we provide an additional formulation of how the maximum entropy RL objective is equivalent to training an amortized approximator of the concept's posterior distribution. We primarily summarize the derivations found in sections 2.2-2.4 of (Levine, 2018) and the appendix of (Korbak et al., 2022) using our notation.

The main deviation is that here we directly show their equivalence as an optimization of an evidence lower bound, given that one wants to optimize over $\log p(o = 1|x)$. That is

$$\log p(o = 1|x) = \log \sum_c p(o = 1, c|x) \tag{37}$$

$$= \log \sum_c p(o = 1|x, c)\pi_0(c|x) \tag{38}$$

$$= \log \left[ \sum_c \pi_\theta(c|x)p(o = 1|x)\frac{\pi_0(c|x)}{\pi_\theta(c|x)} \right] \tag{39}$$

$$\geq \sum_c \pi_\theta(c|x) \log \left[ p(o = 1|x)\frac{\pi_0(c|x)}{\pi_\theta(c|x)} \right] \leftarrow \text{via Jensen's inequality} \tag{40}$$

$$\tag{41}$$

Using the fact that $p(o = 1|x, c) = \exp(r^*(x, c))$ we have :

$$\log p(o = 1|x) \geq \mathbb{E}_{c \sim \pi_\theta} \log \left[ \exp\left(r^*(c, x)\right) \frac{\pi_0(c|x)}{\pi_\theta(c|x)} \right] \tag{42}$$

Therefore, maximizing the above objective is equivalent to:

$$\max_{\pi_\theta} \mathbb{E}_{c \sim \pi_\theta}[r^*(c, x)] - \mathbb{D}_{\text{KL}}(\pi_\theta(c|x) \parallel \pi_0(c|x)) \tag{43}$$

Since optimizing $\mathcal{L}_{\text{CPO}}$ is equivalent to optimizing Equation 43, it is also thereby equivalent to directly optimizing the posterior distribution of the concepts $\pi(c|x)$.

# F. Additional Continual Experiments

Here, we examine the impact of using a model trained with $\mathcal{L}_{\text{BCE}}$ as the starting point for the experiments in § 5.4. Figure 10 compares the performance of CBMs trained on streaming data when initialized with $\mathcal{L}_{\text{BCE}}$ (left) versus $\mathcal{L}_{\text{DPO}}$ (right, same as Figure 7). Overall, we find that updating a $\mathcal{L}_{\text{BCE}}$-initialized model with $\mathcal{L}_{\text{BCE}}$ yields the best results for $\mathcal{L}_{\text{BCE}}$. However, while this improves performance, it still falls short of the results achieved when both initialization and training are done with $\mathcal{L}_{\text{DPO}}$. We note in the leftmost plot, we exclude the $\mathcal{L}_{\text{DPO}}$ model updated without a prior to improve clarity of the plot, but note we find it yields approximately equal results to training with a prior.

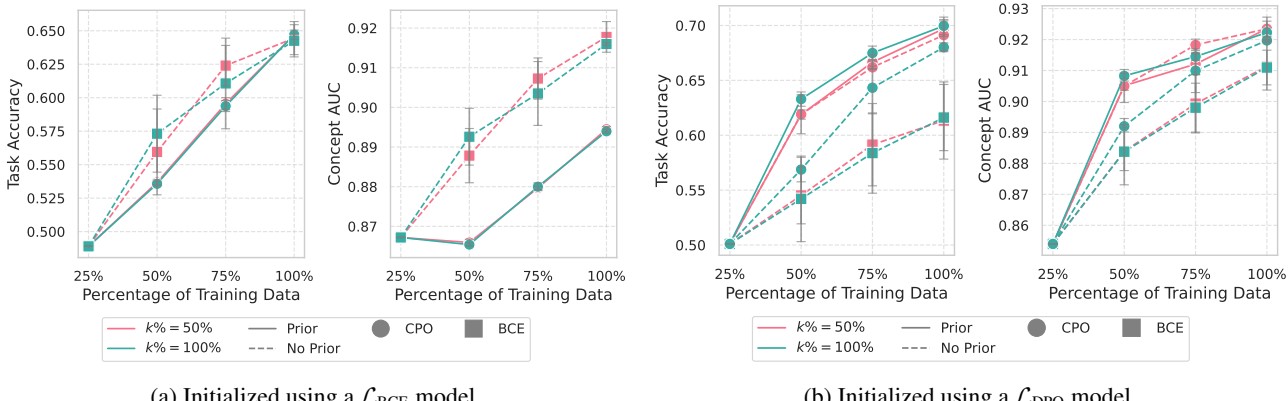

(a) Initialized using a $\mathcal{L}_{\text{BCE}}$ model.

(b) Initialized using a $\mathcal{L}_{\text{DPO}}$ model.

*Figure 10.* Task Accuracy/Concept AUC vs the percentage of data the model has been trained on. We find that updating models initialized with a $\mathcal{L}_{\text{BCE}}$ policy yields improved results for $\mathcal{L}_{\text{BCE}}$ with detrimental ones for $\mathcal{L}_{\text{CPO}}$ (A). In (B) we again visualize the result for updating the models using a $\mathcal{L}_{\text{DPO}}$- initialized policy (same as Figure 7). We find that the best result is given by using $\mathcal{L}_{\text{DPO}}$ to update a $\mathcal{L}_{\text{DPO}}$-initialized policy.

## F.1. Intervention Plots at All Noise Levels

Here we provide an extension of our prior analysis in § 5.2 including intervention performance for all values of $p \in \{0.1, 0.2, 0.3, 0.3\}$. We find that again even at low noise levels $\mathcal{L}_{\text{CPO}}$ models consistently outperform their $\mathcal{L}_{\text{BCE}}$ counterparts. The one model not holding to that is ProbCBMs which outperform $\mathcal{L}_{\text{DPO}}$ consistently on AwA2. We note while this behaviour comes at a cost for ProbCBMs as we show in § 5.2, their concept AUC is severely affected by noise, making the requirement to intervene on these models much harsher.

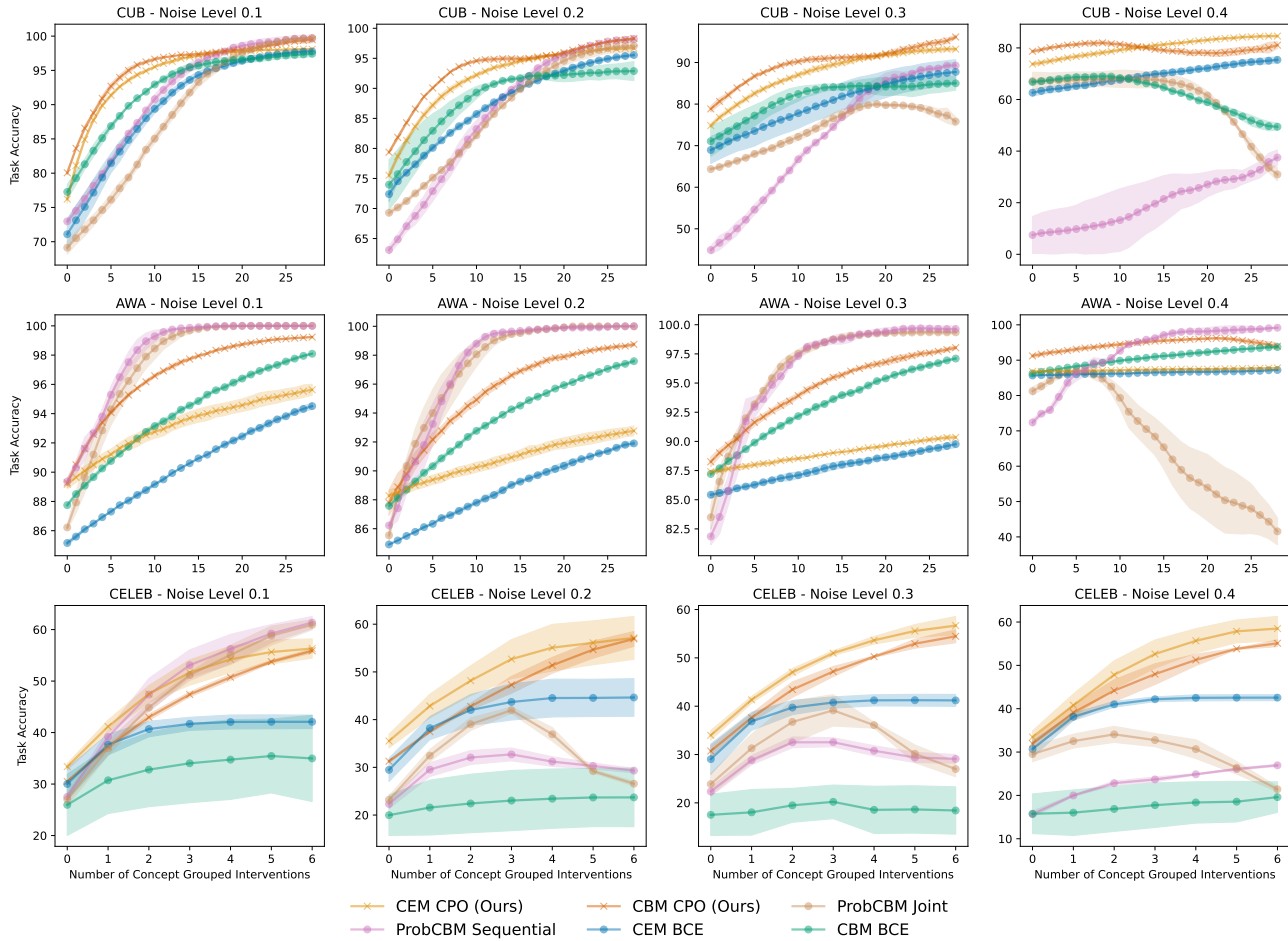

*Figure 11.* Intervention performance for all noise levels. We find that overall methods trained with $\mathcal{L}_{\text{DPO}}$ yield better intervention performance under noise. These findings are specifically relevant to CUB and Celeb where we see all other methods are harshly impacted. We find the only model to consistently outperform $\mathcal{L}_{\text{CPO}}$ are ProbCBMs on AwA2, but have comes at the cost of having their initial concept AUC and task performance severely affected requiring practitioners to intervene far more often.

## F.2. Full Evaluation

In addition to comparing vanilla CBMs and CEMS, Figure 12 provides the full restulst including both joint and sequential ProbCBMs under noise. We generally find that ProbCBMs are extremely senstive to noisy data with their performance cut by half in some metrics such as CUB concept AUC. We find that the uncertainty estimates provided by $\mathcal{L}_{\text{CPO}}$ not only yield better gains in intervention, but also comes with the property of being much more robust to noise.

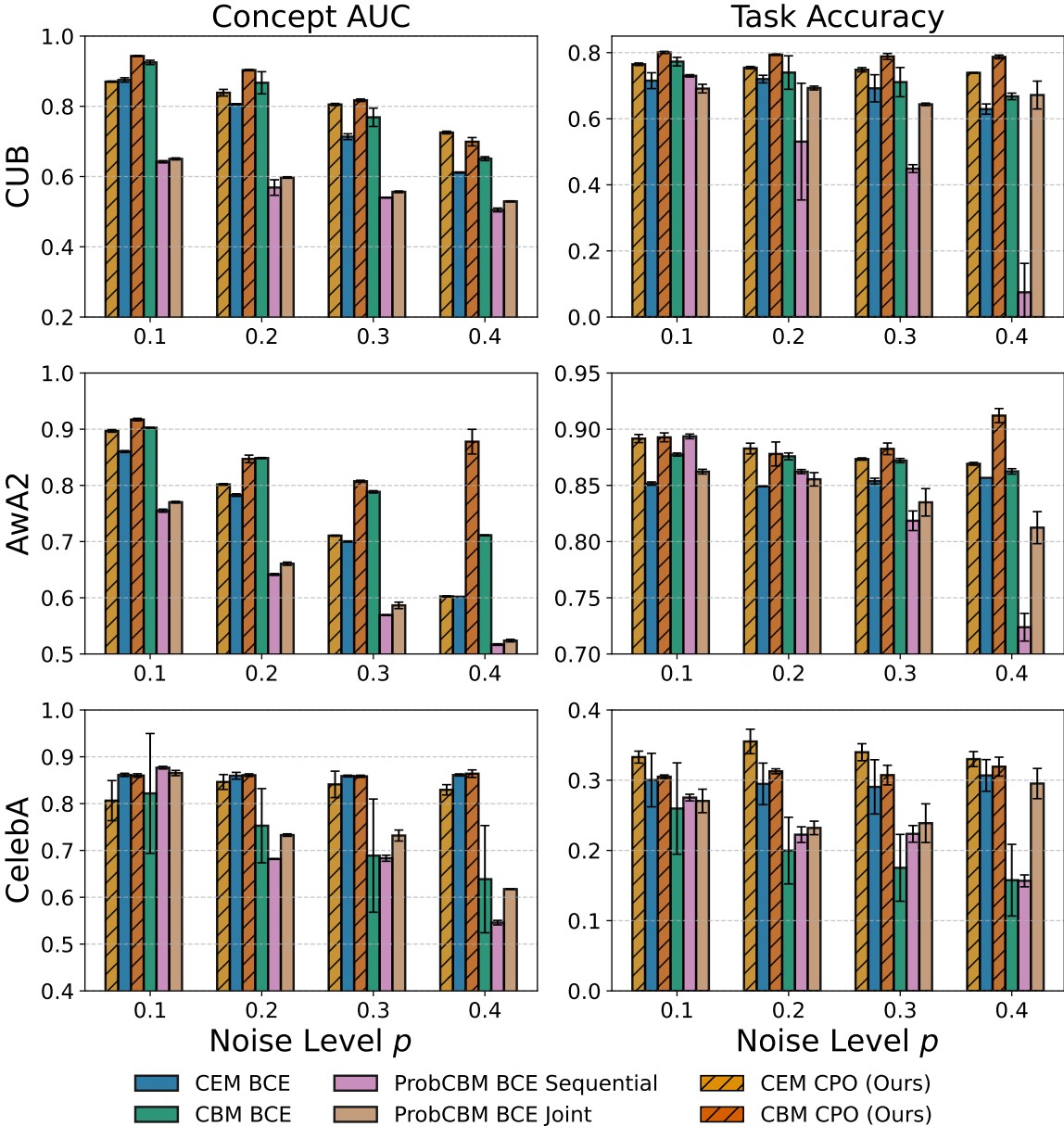

*Figure 12.* Performance metrics for all tasks across varying degrees of label noise. We find that across all noise levels, $\mathcal{L}_{\text{CPO}}$ ourperforms all other models, including both joint and sequential ProbCBMs

# G. Additional Analysis on Uncertainty

In this Appendix, we provide a more in depth analysis of how different CBM models handle uncertainty estimation. We focus on the CUB dataset and visually inspect the baseline uncertainty estimate of various sampled images, observe how different data augmentations affect the uncertainty and quantitatively evaluate how each model's uncertainty changes when the target bird is randomly obscured.

## G.1. Uncertainty Analysis

We display randomly sampled images from the test set. Figure 13 shows these images along with their corresponding uncertainty scores. In this and all subsequent experiments, we define uncertainty as the scaled *variance* of the model's certainty. For ProbCBM, this corresponds to the product of the diagonal entries in the covariance matrix, following (Kim et al., 2023). For $\mathcal{L}_{\text{CPO}}$ and $\mathcal{L}_{\text{BCE}}$, uncertainty is computed as the variance of a Bernoulli random variable, $\sigma_{\text{CPO/BCE}} = c(1-c)$. To enable fair comparison across models, we scale all variance values to lie within the range $[0, 1]$.

For each image, we highlight the concept for which the model exhibits the highest uncertainty and report the uncertainty scores from each model. We observe that $\mathcal{L}_{\text{CPO}}$ tends to report higher uncertainty across most concepts, particularly when the concept in the image is ambiguous or unclear. In contrast, ProbCBM often reports low uncertainty even for visually uncertain concepts. While these examples provide useful intuition, they are limited in scope. In the following experiment, we provide a more thorough quantitative analysis of model uncertainty.

## G.2. Uncertainty Under Augmentations

A standard practice in training vision models (and consequently CBMs) is to apply augmentations during training. These augmentations can obstruct or alter the way concepts are represented in images. To assess whether $\mathcal{L}_{\text{CPO}}$ more reliably models uncertainty under such conditions, we evaluate model behavior under two augmentations: cropping and obscuring. Figure 14 shows examples of images under these transformations. Interestingly, we find that zooming in (via cropping) often increases model confidence in concept predictions. In particular, $\mathcal{L}_{\text{CPO}}$ frequently exhibits a larger boost in confidence compared to other models. Conversely, when the object in the image is obscured, $\mathcal{L}_{\text{CPO}}$ generally becomes much more uncertain, often reaching the maximum uncertainty score when the concept is fully blocked. This contrasts with other models, which tend to remain overconfident in the presence of the concept, even when it is visually occluded.

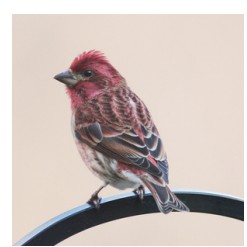

BCE: Has wing pattern = multi-colored (Label:0.0)
U: BCE = 0.98 (P: 0.58) | CPO = 1.00 (P: 0.47) | ProbCBM = 0.88 (P: 0.00)

CPO: Has bill color = black (Label:0.0)
U: BCE = 0.48 (P: 0.14) | CPO = 1.00 (P: 0.48) | ProbCBM = 0.78 (P: 0.00)

ProbCBM: Has breast color = white (Label:0.0)
U: BCE = 0.02 (P: 0.01) | CPO = 0.88 (P: 0.33) | ProbCBM = 0.85 (P: 0.82)

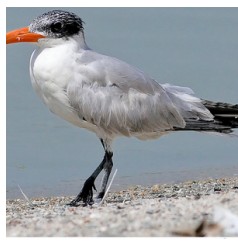

BCE: Has forehead color = white (Label:0.0)
U: BCE = 0.96 (P: 0.60) | CPO = 1.00 (P: 0.47) | ProbCBM = 0.49 (P: 0.85)

CPO: Has back color = grey (Label:0.0)
U: BCE = 0.76 (P: 0.26) | CPO = 1.00 (P: 0.49) | ProbCBM = 0.77 (P: 0.90)

ProbCBM: Has back pattern = solid (Label:1.0)
U: BCE = 0.13 (P: 0.97) | CPO = 0.95 (P: 0.61) | ProbCBM = 0.69 (P: 0.49)

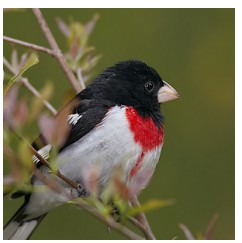

BCE: Has upperparts color = white (Label:1.0)
U: BCE = 0.93 (P: 0.64) | CPO = 0.77 (P: 0.74) | ProbCBM = 0.94 (P: 0.74)

CPO: Has breast pattern = striped (Label:0.0)
U: BCE = 0.00 (P: 0.00) | CPO = 0.95 (P: 0.39) | ProbCBM = 1.00 (P: 0.00)

ProbCBM: Has wing shape = rounded-wings (Label:1.0)
U: BCE = 0.62 (P: 0.19) | CPO = 0.39 (P: 0.89) | ProbCBM = 0.93 (P: 0.51)

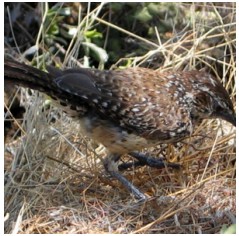

BCE: Has leg color = buff (Label:1.0)
U: BCE = 1.00 (P: 0.47) | CPO = 0.88 (P: 0.67) | ProbCBM = 0.80 (P: 0.99)

CPO: Has bill color = grey (Label:1.0)
U: BCE = 0.28 (P: 0.08) | CPO = 1.00 (P: 0.50) | ProbCBM = 0.00 (P: 0.00)

ProbCBM: Has back color = black (Label:1.0)
U: BCE = 0.62 (P: 0.19) | CPO = 0.99 (P: 0.45) | ProbCBM = 0.26 (P: 0.93)

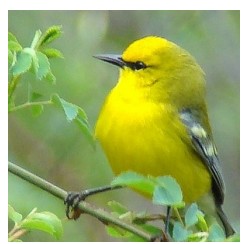

BCE: Has tail pattern = multi-colored (Label:1.0)
U: BCE = 1.00 (P: 0.50) | CPO = 0.93 (P: 0.63) | ProbCBM = 0.00 (P: 0.00)

CPO: Has head pattern = plain (Label:0.0)
U: BCE = 0.82 (P: 0.29) | CPO = 1.00 (P: 0.49) | ProbCBM = 0.00 (P: 0.00)

ProbCBM: Has wing color = yellow (Label:1.0)
U: BCE = 0.63 (P: 0.80) | CPO = 0.91 (P: 0.65) | ProbCBM = 0.90 (P: 0.68)

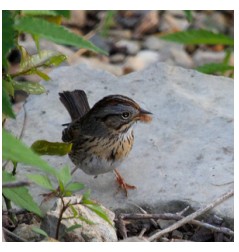

BCE: Has belly color = white (Label:0.0)
U: BCE = 1.00 (P: 0.49) | CPO = 0.99 (P: 0.55) | ProbCBM = 0.37 (P: 0.00)

CPO: Has size = medium (9 - 16 in) (Label:0.0)
U: BCE = 0.01 (P: 0.00) | CPO = 1.00 (P: 0.50) | ProbCBM = 0.75 (P: 0.00)

ProbCBM: Has throat color = grey (Label:1.0)
U: BCE = 1.00 (P: 0.51) | CPO = 0.96 (P: 0.60) | ProbCBM = 0.49 (P: 0.58)

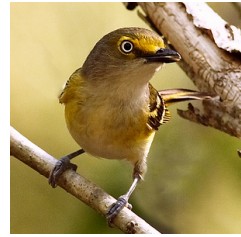

BCE: Has leg color = grey (Label:1.0)
U: BCE = 1.00 (P: 0.50) | CPO = 0.94 (P: 0.62) | ProbCBM = 0.09 (P: 0.00)

CPO: Has tail pattern = multi-colored (Label:1.0)
U: BCE = 0.27 (P: 0.07) | CPO = 1.00 (P: 0.50) | ProbCBM = 0.50 (P: 0.00)

ProbCBM: Has belly color = yellow (Label:1.0)
U: BCE = 1.00 (P: 0.48) | CPO = 0.98 (P: 0.57) | ProbCBM = 0.49 (P: 0.55)

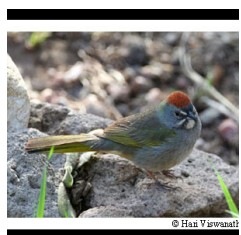

BCE: Has upperparts color = grey (Label:1.0)
U: BCE = 1.00 (P: 0.52) | CPO = 0.93 (P: 0.63) | ProbCBM = 0.64 (P: 1.00)

CPO: Has back pattern = solid (Label:0.0)
U: BCE = 0.86 (P: 0.31) | CPO = 1.00 (P: 0.50) | ProbCBM = 0.61 (P: 0.00)

ProbCBM: Has shape = perching-like (Label:1.0)
U: BCE = 0.83 (P: 0.70) | CPO = 0.97 (P: 0.41) | ProbCBM = 0.56 (P: 0.42)

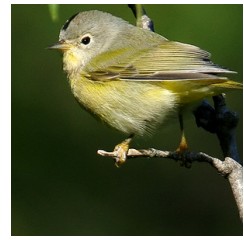

BCE: Has upperparts color = grey (Label:1.0)
U: BCE = 0.98 (P: 0.56) | CPO = 1.00 (P: 0.49) | ProbCBM = 0.79 (P: 1.00)

CPO: Has throat color = yellow (Label:1.0)
U: BCE = 0.45 (P: 0.13) | CPO = 1.00 (P: 0.50) | ProbCBM = 0.57 (P: 0.99)

ProbCBM: Has underparts color = white (Label:0.0)
U: BCE = 0.94 (P: 0.38) | CPO = 0.97 (P: 0.42) | ProbCBM = 0.54 (P: 0.13)

*Figure 13.* Visualization of different images and concept uncertainty score. We display for each model the concept with the highest uncertainty alongside the label for that concept. For each model we display the uncertainty score alongside the probability of the concept being active given by each model. We find that $\mathcal{L}_{\text{CPO}}$ generally has higher uncertainty for more ambiguous concepts and is more certain than other concepts for concepts that are clearly visible.

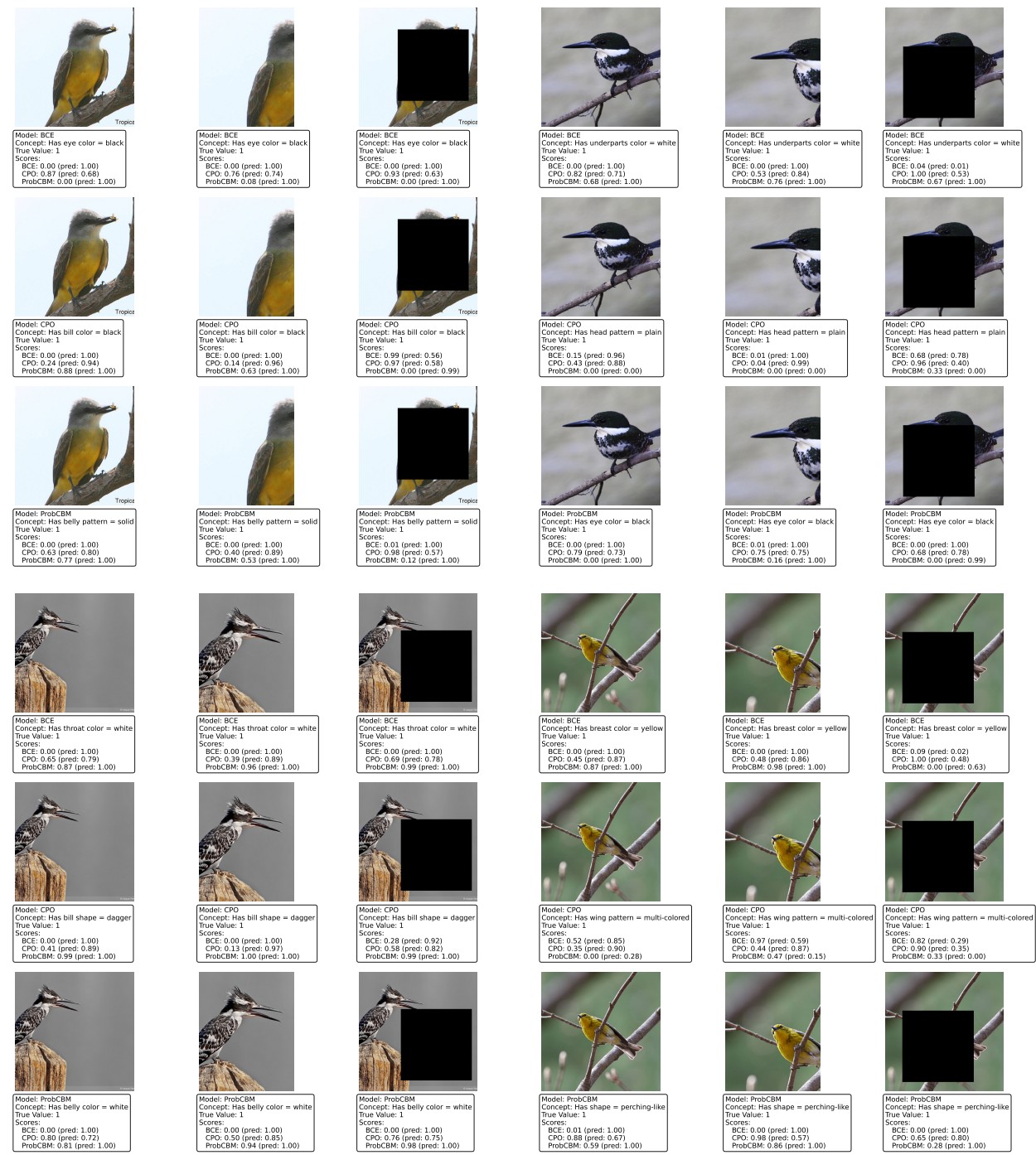

*Figure 14.* Grid visualization of different images before and after applying a cropping and blocking augmentations. For each model we display the concept with the highest certainty score alongside the scores of other models. Overall we find that $\mathcal{L}_{\text{CPO}}$ increases the confidence of the score the most when cropping causes a zooming effect and decreases the most when blocking the target object.

### G.3. Quantitative Uncertainty Analysis

While we previously presented examples where $\mathcal{L}_{\text{CPO}}$ offers improved uncertainty quantification, we now aim to provide a more systematic evaluation. To do this, we apply random blocking augmentations across the entire training set of CUB images. These augmentations are designed to obscure parts of the image and should, in general, increase concept uncertainty

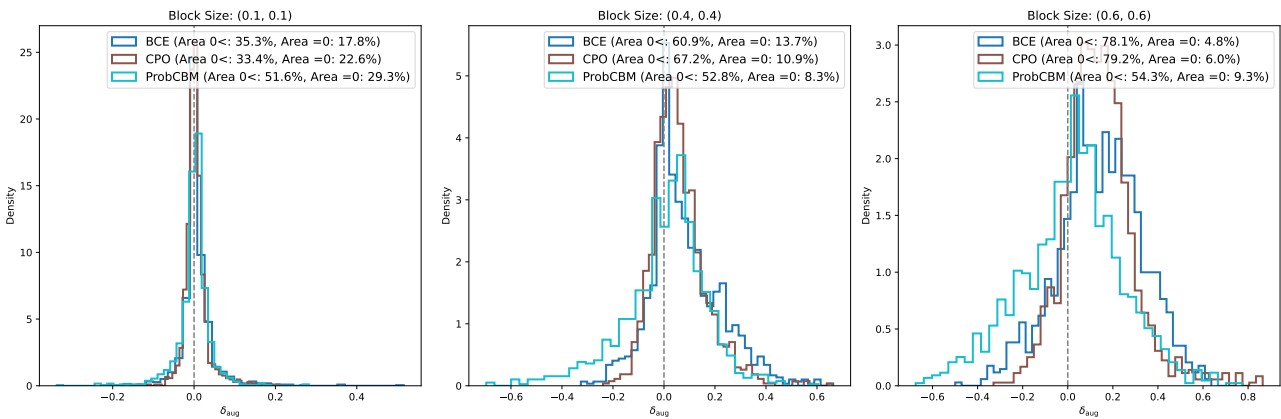

*Figure 15.* Distribution of changes in certainty ($\delta_{\mathrm{aug}}$) for correct concepts under increasing block sizes. Higher area under the curve above zero indicates a more appropriate increase in uncertainty. $\mathcal{L}_{\mathrm{CPO}}$ and $\mathcal{L}_{\mathrm{BCE}}$ respond more consistently to occlusion than ProbCBM, whose uncertainty remains relatively unchanged.

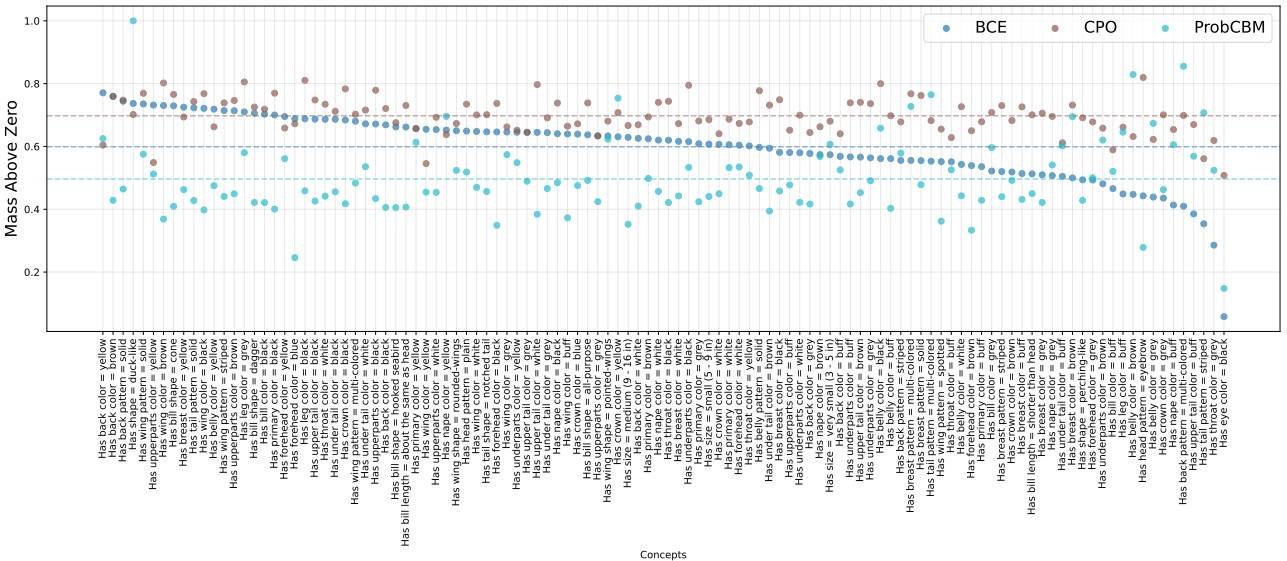

*Figure 16.* Per-concept AUC values under $4 \times 4$ blocking. $\mathcal{L}_{\mathrm{BCE}}$ concentrates uncertainty increases on a few concepts, while $\mathcal{L}_{\mathrm{CPO}}$ distributes uncertainty more evenly across concepts, suggesting better alignment with the effects of occlusion.

by reducing visual access to relevant features. We vary the blocking size, with larger blocks expected to result in greater uncertainty due to a higher likelihood of obstructing the object.

Figure 15 shows the distribution of $\delta_{\mathrm{aug}}$, defined as the change in certainty for *correct concepts* before and after blocking, across three different blocking sizes. We report the area under the curve above zero—higher values indicate that the model tends to become more uncertain after blocking, as desired. Overall, we find that both $\mathcal{L}_{\mathrm{CPO}}$ and $\mathcal{L}_{\mathrm{BCE}}$ increase uncertainty more consistently following augmentation. In contrast, ProbCBM's uncertainty appears largely unaffected by the occlusion, with its scores fluctuating in a seemingly random manner regardless of whether the object is visible.

While $\mathcal{L}_{\mathrm{CPO}}$ tends to yield better uncertainty estimates—particularly at the $0.4 \times 0.4$ blocking size—the differences between $\mathcal{L}_{\mathrm{CPO}}$ and $\mathcal{L}_{\mathrm{BCE}}$ remain modest in aggregate. To better understand the nature of these uncertainty shifts, we examine how they are distributed across individual concepts. Figure 16 presents the per-concept AUC values under $4 \times 4$ blocking. We find that $\mathcal{L}_{\mathrm{BCE}}$ concentrates its uncertainty increases on a small subset of concepts, leaving many concepts largely unaffected by the augmentation. In contrast, $\mathcal{L}_{\mathrm{CPO}}$ exhibits a more balanced distribution, suggesting it more evenly attributes uncertainty

increases across concepts, thus better capturing the effect of occlusion.

## H. Concept Group Noise

A straightforward way to introduce structured noise across datasets is to flip labels between related concepts. This approach mirrors realistic labeling errors, such as a human annotator labeling a bird as having *brown wings* instead of *red* due to factors like lighting, zoom, or viewing angle. We explore this form of structured noise in both the CUB and AwA2 datasets. Following our previous setup, we examine four noise levels: $p \in \{0.1, 0.2, 0.3, 0.4\}$ where $p$ here refers to the probability of having a concept group's label be altered.

The results found in Table 3 are consistent with our earlier findings, where we observe that $\mathcal{L}_{\text{CPO}}$ is significantly less robust to noise compared to $\mathcal{L}_{\text{BCE}}$. Specifically, on the CUB dataset, models trained with $\mathcal{L}_{\text{CPO}}$ achieve substantially higher task and concept accuracies than their $\mathcal{L}_{\text{BCE}}$-based counterparts. In the AwA2 dataset, $\mathcal{L}_{\text{CPO}}$-based models better preserve concept AUC; for example, CBM models trained with $\mathcal{L}_{\text{CPO}}$ consistently maintain concept AUC above $0.84$, while all other models drop below $0.8$. These results suggest that $\mathcal{L}_{\text{CPO}}$ performs better not only under uniform noise but also in the more challenging setting of structured noise.

| Noise Level | | CUB | | AwA2 | |
|---|---|---|---|---|---|
| | | Task Accuracy | Concept AUC | Task Accuracy | Concept AUC |
| $p = 0.1$ | CBM BCE | $0.741 \pm 0.047$ | $0.914 \pm 0.020$ | $0.879 \pm 0.003$ | $0.922 \pm 0.002$ |
| | CBM CPO | $\mathbf{0.796} \pm 0.005$ | $\mathbf{0.941} \pm 0.005$ | $\underline{0.894} \pm 0.003$ | $\mathbf{0.943} \pm 0.003$ |
| | CEM BCE | $0.728 \pm 0.039$ | $0.888 \pm 0.016$ | $0.886 \pm 0.004$ | $0.937 \pm 0.006$ |
| | CEM CPO | $\underline{0.766} \pm 0.003$ | $\underline{0.875} \pm 0.012$ | $\mathbf{0.895} \pm 0.001$ | $\underline{0.931} \pm 0.003$ |
| $p = 0.2$ | CBM BCE | $0.719 \pm 0.039$ | $0.851 \pm 0.018$ | $0.880 \pm 0.001$ | $0.877 \pm 0.001$ |
| | CBM CPO | $\mathbf{0.793} \pm 0.005$ | $\mathbf{0.893} \pm 0.006$ | $\mathbf{0.889} \pm 0.006$ | $\mathbf{0.901} \pm 0.003$ |
| | CEM BCE | $0.674 \pm 0.027$ | $0.796 \pm 0.011$ | $0.884 \pm 0.013$ | $\underline{0.896} \pm 0.002$ |
| | CEM CPO | $\underline{0.757} \pm 0.002$ | $\underline{0.844} \pm 0.002$ | $\underline{0.887} \pm 0.004$ | $0.873 \pm 0.009$ |
| $p = 0.3$ | CBM BCE | $0.681 \pm 0.027$ | $0.757 \pm 0.010$ | $0.880 \pm 0.002$ | $\underline{0.815} \pm 0.002$ |
| | CBM CPO | $\mathbf{0.784} \pm 0.012$ | $\mathbf{0.804} \pm 0.007$ | $\mathbf{0.883} \pm 0.005$ | $\mathbf{0.845} \pm 0.018$ |
| | CEM BCE | $0.671 \pm 0.002$ | $0.704 \pm 0.005$ | $0.864 \pm 0.005$ | $0.735 \pm 0.006$ |
| | CEM CPO | $\underline{0.744} \pm 0.002$ | $\underline{0.809} \pm 0.003$ | $\underline{0.882} \pm 0.002$ | $0.772 \pm 0.010$ |
| $p = 0.4$ | CBM BCE | $0.682 \pm 0.029$ | $0.652 \pm 0.006$ | $\underline{0.878} \pm 0.003$ | $\underline{0.735} \pm 0.004$ |
| | CBM CPO | $\mathbf{0.780} \pm 0.013$ | $\mathbf{0.695} \pm 0.006$ | $0.877 \pm 0.003$ | $\mathbf{0.889} \pm 0.003$ |
| | CEM BCE | $0.667 \pm 0.021$ | $0.610 \pm 0.009$ | $0.870 \pm 0.012$ | $0.652 \pm 0.007$ |
| | CEM CPO | $\underline{0.736} \pm 0.005$ | $\underline{0.715} \pm 0.003$ | $\mathbf{0.883} \pm 0.005$ | $0.685 \pm 0.008$ |

*Table 3.* Task Accuracy and Concept AUC for noising group groups at $p \in \{0.1, 0.2, 0.3, 0.4\}$. We find that like our prior results, $\mathcal{L}_{\text{CPO}}$ significantly aids with concept noise across both CUB and AwA2 datasets.

## I. Computational Analysis

Previously, we noted that $\mathcal{L}_{\text{CPO}}$ adds minimal computational overhead to CBM-based models. Here, we provide a quantitative analysis of the additional compute it requires. Table 4 shows the average training time per epoch using a single RTX-4800 GPU (the same setup used for all reported experiments). We find that $\mathcal{L}_{\text{CPO}}$ adds a negligible amount of additional time per epoch—approximately $0.05$ minutes. This additional clock time is significantly lower than that of other models such as CEM or ProbCBM, which roughly double the training time.

Furthermore, we use PyTorch's built-in implementation of $\mathcal{L}_{\text{BCE}}$, which is optimized for performance, whereas our $\mathcal{L}_{\text{CPO}}$ implementation is hand-written and unoptimized. This suggests that further optimization of $\mathcal{L}_{\text{CPO}}$ could reduce the already

small performance gap even more.

| Model | Avg. Minutes per Epoch |
|---|---|
| CBM BCE | $1.132 \pm 0.012$ |
| CBM CPO | $1.185 \pm 0.014$ |
| CEM BCE | $1.403 \pm 0.055$ |
| CEM CPO | $1.448 \pm 0.080$ |
| ProbCBM | $2.310 \pm 0.035$ |

*Table 4.* Average Time per Epoch

Table 5 reports the total number of parameters for each model, showing that $\mathcal{L}_{\text{CPO}}$ does not introduce any additional parameters. In contrast, models like CEM and ProbCBM increase the parameter count significantly, with ProbCBM requiring roughly four times more parameters.

| Model | Trainable Parameters |
|---|---|
| CBM BCE | 21,364,728 |
| CBM CPO | 21,364,728 |
| CEM BCE | 25,743,889 |
| CEM CPO | 25,743,889 |
| ProbCBM | 89,764,996 |

*Table 5.* Trainable Parameters

## J. Sequential CBMs

While in § 5.1 we explored using $\mathcal{L}_{\text{CPO}}$ with joint CBMs, this is not a requirement. Here, we investigate the use of $\mathcal{L}_{\text{CPO}}$ with sequential CBMs. The results in Table 6 demonstrate that $\mathcal{L}_{\text{CPO}}$ outperforms $\mathcal{L}_{\text{BCE}}$ for training sequential CBMs. This trend holds across all datasets, with a particularly notable improvement in task accuracy on CelebA. We hypothesize this is due to the uncertainty estimate coming from $\mathcal{L}_{\text{CPO}}$, which enables more expressive modeling by the decoder.

| Dataset | Category | Task Accuracy | Concept AUC |
|---|---|---|---|
| CUB | SEQCBM BCE | $0.730 \pm 0.007$ | $0.928 \pm 0.003$ |
| | SEQCBM CPO | $\mathbf{0.741} \pm 0.003$ | $\mathbf{0.931} \pm 0.002$ |
| AWA2 | SEQCBM BCE | $0.898 \pm 0.005$ | $\mathbf{0.964} \pm 0.000$ |
| | SEQCBM CPO | $\mathbf{0.906} \pm 0.001$ | $\mathbf{0.964} \pm 0.000$ |
| CELEB | SEQCBM BCE | $0.297 \pm 0.019$ | $0.879 \pm 0.001$ |
| | SEQCBM CPO | $\mathbf{0.325} \pm 0.007$ | $\mathbf{0.880} \pm 0.000$ |

*Table 6.* Task Accuracy and concept AUC of Sequential CBMs using $\mathcal{L}_{\text{CPO}}$ and $\mathcal{L}_{\text{BCE}}$ across all datasets. Consistent to joint CBMs, we find $\mathcal{L}_{\text{CPO}}$ can increase the unnoised performance. We hypothesize this is due to improved uncertainty estimates of the concepts.

