# OpenReview forum: "Addressing Concept Mislabeling in Concept Bottleneck Models Through Preference Optimization"
_ICML.cc/2025/Conference — ICML 2025 poster_

### Official Review · Reviewer_dJqj · 2025-03-06

**Overall Recommendation:** 3

**Summary:**

CBMs aim to improve explainability of models by making decisions based on human-interpretable concepts but often suffer from mislabeled concept data, leading to significant performance drops. To address this, the Concept Preference Optimization (CPO) objective is introduced, leveraging Direct Preference Optimization to reduce the impact of concept mislabeling. The authors compared CPO  to conventional BCE across multiple datasets to evaluate robustness against concept noise.

**Claims And Evidence:**

The paper claims that CPO improves the robustness of CBMs by replacing correctness assumptions with a preference-based optimization approach. Theoretical results and empirical findings suggest that CPO leads to better performance under noisy concept labels than traditional BCE. The authors provide mathematical derivations to show that CPO's gradient updates remain closer to the optimal noise-free gradient than BCE's, which serves as the main theoretical justification for CPO's robustness. However, while the claims are well-motivated and backed by theoretical derivations, the scope of noise modeling is limited to empirical concept label noise and does not consider more complex, structured noise models (e.g., systematic bias or adversarial noise). Additional empirical results exploring these settings would strengthen the evidence.

**Essential References Not Discussed:**

Thorough analysis.

**Experimental Designs Or Analyses:**

The experimental setup appears reasonable. See below for more.

**Methods And Evaluation Criteria:**

Yes.

**Other Comments Or Suggestions:**

- Table 1 does not follow the order of baseline introduction. Please align the presentation for clarity.
- Consider visualizing gradient behavior over time. A plot comparing LCPO and BCE gradients over training iterations could illustrate the robustness claim more effectively.

**Other Strengths And Weaknesses:**

Strengths:
- The paper presents a novel preference-based optimization approach that relaxes strict correctness assumptions, making it highly relevant for noisy real-world settings.
- Theoretical results are clearly derived, and mathematical proofs support key claims.
- The empirical study demonstrates improvements on multiple real-world datasets, suggesting that CPO has practical benefits.

Weaknesses:
- Strong assumptions (e.g., uniform priors, concept independence) may limit generalizability.
- No discussion of computational efficiency—CPO introduces an online learning approach, but the paper does not analyze how training time or model complexity compares to BCE.
- Limited evaluation on different noise types—structured or adversarial noise is not explored.

**Questions For Authors:**

- Why did the authors only evaluate the jointly trained CBM and not the sequential one?
- What happens when empirical data is systematically biased? The paper assumes empirical preferences are more reliable than random sampling, but what if systematic annotation errors exist (e.g., domain shifts, adversarial perturbations)?
- How would you model concept dependencies (e.g., "wings" and "feathers" in birds) ? The assumption of conditional independence between concepts is unrealistic. How could correlated concept structures be incorporated into CPO?
- How does LCPO perform in early training phases? Proposition 4.2 suggests LCPO’s gradient updates are more conservative than BCE’s, potentially leading to slower convergence. Could adaptive learning rates (scaling updates based on entropy) address this?
- How does CPO handle dataset biases? Since LCPO only modifies the policy when incorrect concepts are sampled, it assumes mislabeled concepts are uniformly distributed. However, in practice, certain incorrect concepts may dominate due to dataset biases. Would reweighting techniques help mitigate this issue?
- Provide a computational efficiency analysis. Given that CPO involves online preference updates, how does its training time and memory usage compare to BCE?

**Relation To Broader Scientific Literature:**

Important topic.

**Theoretical Claims:**

The primary theoretical claim is that CPO is more resilient to noise than BCE due to its gradient properties, specifically, that LCPO’s gradient updates are closer to the optimal noise-free gradient than BCE’s under constant noise conditions. The authors prove this with Proposition 4.1 and Theorem 4.3, which mathematically formalize this gradient similarity. The main limitation here is that the assumptions behind these proofs are strong and may not hold in all real-world cases. First, this assumption of independent and uniform priors for concepts is unrealistic. Many concepts are correlated (e.g., "feathers" and "wings" in birds). Second the noise model assumes random label corruption but does not consider structured noise, such as systematic bias in datasets or adversarial noise attacks. A stronger theoretical contribution could involve relaxing these assumptions and extending the analysis to more realistic noise distributions.

---

> ### Author Rebuttal · Authors · 2025-04-01
>
> We thank the reviewer for their feedback and helping us improve our work.
>
> `Consider visualizing gradient behavior over time...`
>
> We agree with the reviewer that this figure can aid in illustrating our claim. A version of it is already in App D, where we empirically confirm our theoretical results.
>
> `1 Sequential CBMS`
>
> We chose to focus on joint CBMs as they are shown to generally be the best-performing models for a variety of tasks. Below, we provide some results for the unnoised setting using sequential CBMs. We generally observe that CPO provides better results for both the sequential and joint setting, specifically in task accuracy, where we find that the added uncertainty estimates encoded into the concept representation allows for more expressive modeling to be picked up by the decoder.
>
> Sequential CBM results: https://ibb.co/bMJKXcLP
>
> ` 2. What happens when empirical data is systematically biased?`
>
> 1.1  We agree that incorporating experiments with more structured noise would strengthen our conclusions. Specifically, in our response to reviewer **sJF1** we describe the following experiments:
>
> One where noise is introduced at the concept group level (e.g., flipping only wing-related attributes in birds or altering beak colors).
>
> Another where noise is introduced based on the confidence level provided by the labeler (as available in the CUB dataset).
>
> ` 3.Strong assumptions`
>
> While our results do rely on the assumption of a uniform prior, this is the weakest prior one can have. We show in Sec 5.3 how using a prior improves the performance of the model through the added flexibility. This ability to leverage a prior (which BCE cannot) but still work using an uninformative one is beneficial. Thus, we believe that while it is an assumption that our results rely on, it is not impairing.
>
> While we acknowledge that concepts are often correlated, assuming conditional independence is a common approach in CBM literature, with dedicated works exploring dependency modeling (e.g., stochastic CBMs [1]). Given this, we opted for a model that assumes conditional independence.
>
> This feedback has prompted us to reflect on whether conditional independence is necessary for the correctness of our analysis. Upon review, we found that it is not required, meaning our results can generalize to more expressive models.
>
> To illustrate this, consider modeling the joint distribution of a concept with all other concepts, $\pi(c_i, c_{0:i-1}|x)$ . Using the chain rule, this can be factorized into an autoregressive formulation:
>
> $$
> \pi(c_i | c_{i-1}, x)\pi(c_{i-1} | c_{i-2}, x) \dots
> $$
>
> With this factorization, we can always compute the joint distributions $\pi(c | x)$ and $\pi(c' | x)$, ensuring that our results in Appendix C.1 still hold even without only assuming conditional independence on x.
>
> The main reason for the independence assumption was to simplify the derivation of the CPO loss in Eq. (6). Since we find this assumption unnecessary, we will drop this assumption from our work.
>
> Here, we present preliminary results using this autoregressive formulation. Specifically, we implement this approach with two networks: $ \pi_{\phi}(c_{i} \mid c_{0:i-1}, x) $ and $ h_{\omega}(\tau \mid c, x) $, where $ h_{\omega} $ determines the order of the autoregressive decomposition.  In this process, $h_{\omega}(\tau \mid c, x) $ takes as input the concept probabilities predicted by $\pi $ and produces a softmax distribution over concepts. The concept with the highest probability is then selected and predicted by $\pi $, and this procedure repeats in an autoregressive manner until all concept probabilities have been obtained.
>
> AR CBM Table: https://ibb.co/BH9s5jcm
>
> Here, we find that our autoregressive model beats conditionally independent models for BCE, slightly underperforming for CPO. Regardless, we find that CPO also performs better for this type of model.
>
> ` 4. How does LCPO perform in early training phases? Proposition 4.2 suggests LCPO’s gradient updates are more conservative than BCE’s, potentially leading to slower convergence...`
>
> In practice, CPO can be slower in fitting training data, but we observe that this does not negatively impact generalization. Instead, it suggests that CPO may be less prone to overfitting, which is a known issue in likelihood-based training. To illustrate this, we provide anonymized links for sample runs on the CUB dataset across two seeds.
>
> https://ibb.co/Z67qYySF (Validation Task Accuracy)
>
> https://ibb.co/4wkNn1WW (Validation C AUC)
>
> https://ibb.co/Xx8WNLnZ (Training task accuracy)
>
> https://ibb.co/chxX81Jf (Training C AUC)
>
> Regarding adaptive learning rates or entropy-based updates, any improvements seen from these strategies should also apply to CPO, as its updates are fundamentally based on data likelihood.
>
> `Computational Analysis`
>
> Please see our response to reviewer **sJF1**
>
>
> [1] Stochastic Concept Bottleneck Models  https://arxiv.org/html/2406.19272v1

---

### Official Review · Reviewer_3SWh · 2025-03-13

**Overall Recommendation:** 4

**Summary:**

One limitation of concept bottleneck models (CBMs) is that their training requires the set of correct concept annotations for all samples. However, concept mislabelling is inevitable due to labeling noise or subjective annotations. To this end, this paper proposes a CBM that is robust to concept-label noise. Specifically, inspired by the recent progress in Preference Optimization, Direct Preference Optimization (DPO) in particular, they introduce a novel objective function for training CBMs, named Concept Preference Optimization (CPO). Theoretically, they show the similarities and differences between the traditional BCE loss and the CPO loss, explaining why the latter is more robust to concept-label noise. In addition, they demonstrate that CPO is equivalent to learning the concept’s posterior distribution. Empirically, they show that CBMs trained with the CPO loss perform better in both un-noised and noisy environments.

## Update after rebuttal
I keep my score since the author's responses have addressed my questions.

**Claims And Evidence:**

The claims made in the submission are supported by clear and convincing theoretical and empirical evidence.

**Essential References Not Discussed:**

None.

**Experimental Designs Or Analyses:**

The experimental designs seem valid to me, because the datasets used in this paper are the standard ones and the experiments have studied two different environments—with and without concept-level noise (even with different levels of noise), which well demonstrate the effectiveness of the proposed method across various scenarios.

**Methods And Evaluation Criteria:**

The datasets used in this paper are the standard benchmarks used to evaluate the performance of CBMs, including CUB, AwA2, and CelebA.

The studied problem (i.e., training a model robust to concept-level noise) is well motivated because there indeed exists much noise in these datasets.

**Other Comments Or Suggestions:**

In Eq (6), $D$ should be $\mu$?

**Other Strengths And Weaknesses:**

This is a solid piece of work, clearly defining the problem, formulating the method, theoretically analyzing the properties of the proposed loss in comparison to the traditional loss, and evaluating its effectiveness across various environments.

**Questions For Authors:**

1. In Eq (5), are $c$ and $c^\prime$ a scalar or a k-dimensional vector?

2. Different from DPO which adopts a static preference dataset, the proposed method CPO adopts a dynamic dataset because $c^\prime$ is sampled from the current model $\pi_\theta$. My question is: I understand that in the early training stage, $c \succ c^\prime$, because $\pi_\theta$ is not well-trained at that time. But, if it is possible that in the late training stage, $c^\prime \succ c$ could happen? If this is the case, will continuing training the model with the proposed loss hurt the model's ability to distinguish preference pairs, i.e., give a higher score to the better concept?

3. Can I conclude the reason why CPO loss is more robust against the concept-level noise as follows: DPO is the same as BCE when $c\sim\mu^{+}$; however, when $c\sim\mu^-$, DPO yields smaller gradients?

4. I am just curious: in the future, if it's possible to utilize CPO to identify mislabeled concepts or even find the correct labels for these concepts?

**Relation To Broader Scientific Literature:**

This paper focuses on the robustness of CBMs against concept-level noise, which is a well-motivated problem and may inspire further research on removing concept-level noise in datasets or building noise-robust CBMs.

This paper designs a loss function based on Direct Preference Optimization (DPO) and may inspire further research on combining the field of preference optimization and the area of concept bottleneck models.

**Theoretical Claims:**

The theoretical claims and proofs seem correct to me.

---

> ### Author Rebuttal · Authors · 2025-04-01
>
> We thank the reviewer for their feedback and encouragements. Here we provide some detailed responses to some of the questions you have raised.
>
> `In Eq (6),  should $D$ be $\mu$ ?`
>
> Yes, that is a notational mistake on our end. We will fix it.
>
> `1. In Eq. (5), are c and c′ scalars or k-dimensional vectors?`
>
> In our work, Eq. (5) uses scalars. However, this is not a strict requirement—one could use vectors to structure concepts in this way. That said, scalars are significantly less computationally demanding, as modeling structured objects would lead to $2^k$ possible concept representations, which could become impractical.
>
> `2. In the later training stages, could c′ ≻ c occur? If so, would continued training hurt the model's ability to distinguish preference pairs?`
>
> Yes, this situation can arise later in training. To mitigate this, we apply early stopping across all models. This ensures that the model does not overfit to later-stage preferences in a way that degrades its ability to distinguish concept quality.
>
> `3. Is the reason CPO is more robust against concept-level noise that DPO behaves like BCE when c ∼ μ+, but for c ∼ μ−, DPO yields smaller gradients?`
>
> This intuition is correct. In **Appendix D**, we show that when noise is low (e.g., **0.1**), CPO’s gradients are indeed smaller than BCE’s. This is consistent with our experiments, where BCE performs reasonably well under **p_noise = 0.1**. However, at **p_noise = 0.3**, CPO’s gradients are **orders of magnitude smaller** than BCE’s. This aligns with our empirical findings—CPO remains relatively unaffected by high noise levels, whereas BCE struggles significantly.
>
> `4. Could CPO be used to identify mislabeled concepts or even recover correct labels?`
>
> This is an interesting question. While we have not explored this explicitly, we have qualitatively found that CPO’s uncertainty estimations seem more grounded to the target object in the image. To show this, we provide examples of images before and after various augmentations, along with the concept being analyzed. Our uncertainty score is based on the variance of the concept, normalized to a [0,1] range, where 0 indicates complete certainty and 1 represents maximum uncertainty. Results show that when the target object becomes obscured, CPO more effectively increases its concept uncertainty, while both BCE and ProbCBM perform significantly worse at this task. Additionally, we observe that in certain scenarios, cropping the image creates a zooming effect that makes models more confident in their assessments.
>
> https://ibb.co/JR9DZvfd
>
> https://ibb.co/219TTtLN
>
> https://ibb.co/23Rkwqqr

---

> > ### Comment · Reviewer_3SWh · 2025-04-02
> >
> > I thank the authors for the responses, which have addressed my questions, so I will keep my score.
> >
> > It would be appreciated if the authors could include A2, i.e., the discussion on $c^{\prime} \succ c$ in the later training stages, in the paper.

---

> > > ### Author Response · Authors · 2025-04-04
> > >
> > > We thank the reviewer for their constructive feedback and will add this discussion as either a footnote in section 4 or to the appendix.

---

### Official Review · Reviewer_sJF1 · 2025-03-13

**Overall Recommendation:** 3

**Summary:**

The paper proposes training CBMs (and its variants) using Concept Preference Optimization (CPO) -  a method directly borrowing from the Preference Optimization (PO) literature

**Claims And Evidence:**

Yes.

**Essential References Not Discussed:**

All references present.

**Experimental Designs Or Analyses:**

Yes, all experiments are standard.

**Methods And Evaluation Criteria:**

Yes.

**Other Comments Or Suggestions:**

Refer Weakness.

**Other Strengths And Weaknesses:**

Strengths:
1. Strong work combining interesting PO approaches with CBMs.
2. Empirical evaluation and theoretical justification are sound.
3. The work is a step towards increasing the robustness of CBMs.

Weakness:
1. Presentation Issues: The paper can significantly benefit from presentation issues. For example, 3 pages into the reading, there is only one motivation example which is not discussed well. I would implore authors to add some concrete examples and motivation in clear sentences to make this a more appealing paper.
2. The experiments seem too synthetic: Even though I understand the motivation of the approach, and the experiment design - the setting is a bit too synthetic for a good analysis. In particular, label flipping with a probability can be one of the experiments, but how about mislabelling *similar* concepts, which is a much more likely scenario? For example - red wings flip to 0, brown wings flip to 1. You can utilize CLIP/Bertscore to do something like that. This will make the experiments much more "life-like".
3. Better presentation of Figure 4: Fig-4, the most important figure in the paper suffers from a lack of visual intuitiveness in my opinion. The bars are just too close together to convey a strong improvement message. Why not just make a Table, like Table-1?

**Questions For Authors:**

1. What exactly is the ground truth concept distribution the preference is calculated against? Is it taken directly from the dataset annotation or is it done manually?
2. What are the computational overheads if any? Are there ANY limitations at all?

**Relation To Broader Scientific Literature:**

The paper studies an important problem of CBM training with a trending topic of PO.

**Theoretical Claims:**

Yes.

---

> ### Author Rebuttal · Authors · 2025-04-01
>
> We sincerely thank the reviewer for their time and effort in helping us improve our work.
> ### W 1
> We agree. We will address these concerns in the camera-ready version. We discuss the details in comment 2 of our response to reviewer **2knp**.
>
> ### W 2
>   We also agree (thank you). As you and other reviewers have pointed out, experiments with more structured noise would strengthen our results. To address this, we have designed two experiments with more structured noise to the concepts than our original setting.
>
> ### Noising by Group Level
> As you suggested, we now study mislabeling similar concepts based on concept groups in the CUB and AwA2 datasets. This means we introduce noise to semantically similar concepts, such as switching *red wings* to *brown wings* (as in your example). In this experiment, we apply noise at the group level with different noise levels,  $p\in$ {$\{0.1,0.2,0.3,0.4\}$}.
>
> Full results are below. We observe that models trained using BCE experience a substantial drop in performance, whereas CPO models remain much more robust. For instance, in CUB task accuracy in CBM CPO remains relatively stable even at high noise levels.  Likewise, in AwA2 we observe CPO models are able to preserve Concept AUC performance much better compared to their BCE counterparts.
>
> Unfortunately, due to character limits in our response, we are restricted to providing links to these results. If any reviewer is uncomfortable clicking the link, we will try to provide a version of the full table in our next (final) response.
>
> CUB and AwA2 Noising by Group:
>
> https://ibb.co/NgDZ9sBC
>
> ### Uncertainty Experiment
> Additionally, we leverage a useful property of the CUB dataset: labelers provide confidence scores for their annotations. These scores range from $\{1,2,3,4\}$, where higher values indicate greater confidence. We introduce noise proportionally to these scores—labels with lower confidence are more likely to be flipped. Specifically, we apply noise at the following rates:
> - Confidence 1 → 40% noise
> - Confidence 2 → 30% noise
> - Confidence 3 → 20% noise
> - Confidence 4 → 10% noise
>
> CUB confidence-based results:
>
> https://ibb.co/KcfB6qHp
>
> As in other experiments, BCE-based models are heavily impacted by noise, leading to significant drops in task accuracy and concept AUC. In contrast, CPO-based models consistently exhibit robustness to noise, regardless of their structure.
>
> We thank all reviewers for suggesting experiments with more structured noise, as we believe these results significantly improve the quality of our work.
>
> ### W 3
> This is a good point; improving visual clarity is important. We are experimenting with converting Figure 4 into a table. However, we initially chose a plot for its space efficiency. At a minimum, we will include a table with all metrics in the appendix for the camera-ready version.
>
> Additionally, if we retain the figure, we may remove Prob-CBMs from the plot to enhance clarity, restricting their results to the appendix given their underperformance in this setting.
>
> # Questions
>
> ` What exactly is the ground truth concept distribution that the preference is calculated against?...`
>
> The preferred concepts are indeed taken from the empirical dataset. We  discuss it in lines 165-170:
>
> *"To circumvent this issue, we can leverage the empirical dataset and state its preference over a concept set sampled from $\pi_\theta$. The preference over a pair of concepts should hold specifically early on in training where the policy is suboptimal compared to the empirical data."*
>
> ` What are the computational overheads, if any...`
>
> We thank the reviewer for suggesting improvements to our computational analysis. Below, we provide a detailed breakdown of the computational overhead associated with each model.  The following table reports the average time per epoch (in minutes) for all models, measured over a full run on the CUB dataset. Additionally, we include the number of trainable parameters for each model.
>
> https://ibb.co/8gL3ZNLq
>
> While CPO introduces a small computational overhead, the increase is minimal—approximately 0.05 minutes per epoch compared to BCE. This is in contrast to CEM and ProbCBM, which significantly increase runtime. Moreover, we compare an unoptimized implementation of CPO against an optimized BCE implementation built specifically for efficiency in PyTorch. This suggests that further optimizations could reduce CPO’s overhead even more.
>
> On the other hand, ProbCBM has 4× the number of parameters compared to CBM and nearly doubles the training time, despite underperforming CPO in most settings.
>
> A potential limitation of CPO is that its gradients tend to be smaller—particularly in the early stages of training—compared to BCE. This could lead to slower convergence in some cases. However, as we also mention in our response to reviewer dJqj, we do not observe this empirically.

---

### Official Review · Reviewer_2knp · 2025-03-14

**Overall Recommendation:** 4

**Summary:**

The paper proposes a Preference Optimization (PO) based training Paradigm - CPO for training CBMs. The paper gives a detailed analysis of the proposed method and has experiments on intervention and random label flips. The preference set is taken as observed empirical evidence while the negative sampling of concepts is taken as unpreffered.

**Claims And Evidence:**

Yes.

**Essential References Not Discussed:**

All references are discussed well.

**Experimental Designs Or Analyses:**

Yes, the experiments are appropriate.

**Methods And Evaluation Criteria:**

Yes.

**Other Comments Or Suggestions:**

Refer weakness

**Other Strengths And Weaknesses:**

Strengths:
1. The paper addresses a well-known but rarely explored aspect of CBMs - concept mislabelling. Usually, concept labels are treated as almost always accurate but can be susceptible to mislabelling.
2. The paper is well-written and easy to follow. The theoretical analysis and equations are clear to understand.


Weakness:
1. Diverse Experiments will strengthen the conclusion: For the "Noised Evaluation" experiment, the experiment setting tests the performance of such models on random concept flips. This evaluation leads to 2 problems - 1) The ideal real-world setting does not directly entail random label flip, but a label-confound, where a user can probably mislabel based on their perception of the image. 2) If the model still performs almost as well when the label flips with 0.4 probability, it is deviating from its empirical evidence and relying more on the posterior. For (2) - should we actually ignore empirical evidence?
2. Confusing Writing: The paper can benefit much more from improving the flow and writing in the initial sections. A dedicated paragraph discussing the example figure in simple English can make the persuasiveness and overall appearance much more appealing. One has to shift to Section-3 for a thorough understanding directly.

**Questions For Authors:**

How are the hyperparameters tuned for the model?

**Relation To Broader Scientific Literature:**

The new CBM training mechanism fixes known problems.

**Theoretical Claims:**

Yes, extensively.

---

> ### Author Rebuttal · Authors · 2025-04-01
>
> We sincerely thank the reviewer for their time and effort in helping us improve our work.
> ## Weaknesses
>
> ### 1 Diverse Experiments to Strengthen Conclusions
>
>
> 1.1 Thank you for this valuable feedback. We agree that incorporating experiments with more structured noise would strengthen our conclusions. In our response to reviewer **sJF1**, we discuss two additional experiments:
>
> One where noise is introduced at the concept group level (e.g., flipping only wing-related attributes in birds or altering beak colors).
>
>
> Another where noise is introduced based on the confidence level provided by the labeler (as available in the CUB dataset).
>
> 1.2
> `
> If the model still performs almost as well when the label flips with 0.4 probability, it is deviating from its empirical evidence and relying more on the posterior. For (2) - should we actually ignore empirical evidence?
> `
>
>
>  While it is possible that the model’s strong performance is partially due to reliance on the posterior rather than empirical evidence, we do not believe this fully explains the results. For example, in some settings, particularly for CEMs trained with BCE, models can still perform well even under high levels of noise.
> We attribute this to the fact that, despite 40% of the labels being noisy, the remaining 60% provide sufficient empirical evidence for the model to fit the data—albeit imperfectly.
>
>
>
> ### 2. Improving the Introduction for Clarity
> We appreciate the reviewer’s suggestion to include a dedicated paragraph explaining Figure 1 in the introduction. While we agree that this addition would improve clarity, space constraints may limit our ability to introduce an entirely new paragraph. Instead, we propose enhancing the introduction by adding context around Figure 1, making it more digestible for the reader.
> Here, we provide the modified text we would add to the introduction to address this issue
> ```
> We propose Concept Preference Optimization (CPO), a policy optimization-inspired objective loss for CBMs. Figure 1 illustrates how CPO leverages pairwise comparisons of concept preferences to guide updates toward preferred concepts while mitigating the impact of incorrect gradients. Unlike traditional likelihood-based learning, which updates on all samples regardless of correctness, CPO selectively adjusts based on sampled preferences. This reduces sensitivity to noise by being able to mitigate incorrect gradient updates when incorrect concepts are sampled. Our analysis shows that CPO is equivalent to learning the posterior distribution over concepts, leading to more robust training. Empirically, we demonstrate that CPO not only improves CBM performance in noise-free settings but also significantly alleviates the impact of concept mislabeling.
> ```
>
> Question
> 1.  How are the hyperparameters tuned for the model?
>
> We discuss these details in Appendix A.1, which we provide here for reference:
>
> ```
> From App A. We use a batch size of 512 for the Celeb dataset and 256 for CUB and AwA2. We train all
> models using RTX8000 Nvidia-GPU. In all datasets, we train for up to 200 epochs and early stop if the validation loss has not improved in 15 epochs. For fair evaluation across methods, we tune the learning rate for CEMs, CBMs, and ProbCBM. Specifically, for CUB and AwA2 datasets, we explore learning rates ∈ {0.1, 0.01}, while for CelebA, we expand the search to ∈ {0.1, 0.01, 0.05, 0.005} due to the observed instability of CEMs at higher learning rates. Additionally, we set the hyper-parameter λ ∈ {1, 5, 10} for all methods. For CEMs and models trained using LDPO, we found RandInt beneficial, which randomly intervenes on 25% of the concepts during training. ProbCBM introduced a few extra hyperparameters, which we did not tune in this work, and we directly used the hyperparameters provided by the original authors. Similar to other models, ProbCBM employs RandInt at 50%, making it particularly sensitive to interventions, especially in concept-complete tasks such as AwA2 and CUB. The only model for which we tune additional hyper-parameters is Coop-CBM, where we adjust the weight parameter for the auxiliary loss.
> ```
>
>
>
> We hope our additional experiments and clarifications can help alleviate your concerns.

---

> > ### Comment · Reviewer_2knp · 2025-04-04
> >
> > I appreciate the experiment regarding group-level noise analysis. Utilizing labeler confidence is an interesting approach and I hope/trust the authors will provide a more detailed analysis of the experiment in the final camera-ready version. The experiment design will help in improving present CBM architectures as well.
> >
> > I am improving my score accordingly.

---

> > > ### Author Response · Authors · 2025-04-08
> > >
> > > We thank the reviewer again for their feedback. We will include a more detailed analysis of both these experiments in the final draft of the paper.

---

### Decision · Program_Chairs · 2025-05-01

**Decision:**

Accept (poster)

**Comment:**

All reviewers agree that this is solid work that combines interesting PO approaches with CBMs.
The work is a step towards increasing the robustness of CBMs, and it would be of interest to the ICML community